# The Value of Reward Lookahead in Reinforcement Learning

**Nadav Merlis**
FairPlay Joint Team, CREST, ENSAE Paris
`nadav.merlis@ensae.fr`

**Dorian Baudry**
FairPlay Joint Team, CREST, ENSAE Paris
Institut Polytechnique de Paris

**Vianney Perchet**
FairPlay Joint Team, CREST, ENSAE Paris
Criteo AI Lab

## Abstract

In reinforcement learning (RL), agents sequentially interact with changing environments while aiming to maximize the obtained rewards. Usually, rewards are observed only *after* acting, and so the goal is to maximize the *expected* cumulative reward. Yet, in many practical settings, reward information is observed in advance – prices are observed before performing transactions; nearby traffic information is partially known; and goals are oftentimes given to agents prior to the interaction. In this work, we aim to quantifiably analyze the value of such future reward information through the lens of *competitive analysis*. In particular, we measure the ratio between the value of standard RL agents and that of agents with partial future-reward lookahead. We characterize the worst-case reward distribution and derive exact ratios for the worst-case reward expectations. Surprisingly, the resulting ratios relate to known quantities in offline RL and reward-free exploration. We further provide tight bounds for the ratio given the worst-case dynamics. Our results cover the full spectrum between observing the immediate rewards before acting to observing all the rewards before the interaction starts.

## 1 Introduction

Reinforcement Learning [RL, Sutton and Barto, 2018] is the problem of learning how to interact with a changing environment. The setting usually consists of two major elements: a transition kernel, which governs how the state of the environment evolves due to the actions of an agent, and a reward given to the agent for performing an action at a given environment state. Agents must decide which actions to perform in order to collect as much reward as possible, taking into account not only the immediate reward gain, but also the long-term effects of actions on the state dynamics.

In the standard RL framework, reward information is usually observed after playing an action, and agents only aim to maximize their cumulative *expected* reward, also known as the *value* [Jaksch et al., 2010, Azar et al., 2017, Jin et al., 2018, Dann et al., 2019, Zanette and Brunskill, 2019, Efroni et al., 2019b, Simchowitz and Jamieson, 2019, Zhang et al., 2021b]. Yet, in many real-world scenarios, partial information about the future reward is accessible in advance. For example, when performing transactions, prices are usually known. In navigation settings, rewards are sometimes associated with traffic, which can be accurately estimated for the near future. In goal-oriented problems [Schaul et al., 2015, Andrychowicz et al., 2017], the location of the goal is oftentimes revealed in advance. This information is completely ignored by agents that maximize the expected reward, even though using this future information on the reward should greatly increase the reward collected by the agent.

As an illustration, consider a driving problem where an agent travels between two locations, aiming to collect as much reward as possible. In one such scenario, rewards are given only when traveling free roads. It would then be reasonable to assume that agents see whether there is traffic before deciding in which way to turn at every intersection ('one-step lookahead'). In an alternative scenario, the agent participates in ride-sharing and gains a reward when picking up a passenger. In this case, agents gain information on nearby passengers along the path, not necessarily just in the closest intersection ('multi-step lookahead'). Finally, the destination might be revealed only at the beginning of the interaction, and reward is only gained when reaching it ('full lookahead'). In all examples, the additional information should be utilized by the agent to increase its collected reward.

In this paper, we analyze the value of future (lookahead) information on the reward that could be obtained by the agent through the lens of competitive analysis. More precisely, we study the *competitive ratio* (CR) between the value of an agent that only has access to reward distributions and that of a lookahead agent who sees the actual reward realizations for several future timesteps before choosing each action. Our contributions are the following: **(i)** Given an environment and its expected rewards, we characterize the distribution that maximizes the value of lookahead agents, for all ranges of lookahead from one step to full lookahead; this distribution therefore minimizes the CR. In particular, we show that the lookahead value is maximized by *long-shot* rewards – very high rewards at extremely low probabilities. **(ii)** We derive the worst-case CR as a function of the dynamics of the environment (that is, for the worst-case reward expectations). Surprisingly, the CR that emerges is closely related to fundamental quantities in reward-free exploration and offline RL [Xie et al., 2022, Al-Marjani et al., 2023]. **(iii)** We analyze the CR for the worst-possible environment. Specifically, tree-like environments that require deciding both *when* and *where* to navigate exhibit near-worst-case CR. **(iv)** Lastly, we complement these results by presenting different environments and their CR, providing more intuition to our results.

**Related Work.** The idea of utilizing lookahead information to update the played policy is related to a control concept called Model Predictive control [MPC, Camacho et al., 2007], also known as receding horizon control. In complex control problems, it could be challenging to predict the system behavior in long horizons due to errors in the model or nonlinear dynamics. To mitigate this, MPC designs a control scheme for much shorter horizons, where the model is approximately accurate, oftentimes on a simplified (e.g., linearized) model. Then, to correct the deviations due to modeling errors, MPC continuously updates the controller according to the actual system state. In our context, the localized system estimates could be seen as lookahead information. Similar ideas have also been used for planning in reinforcement learning settings [Tamar et al., 2017, Efroni et al., 2019a, 2020]. Yet, these concepts are mainly used to improve planning efficiency and account for nonlinearities/disturbances in the model. A few notable exceptions study the competitive ratio (and/or dynamic regret) between controllers with partial lookahead information to ones with full information [Li et al., 2019, Zhang et al., 2021a, Lin et al., 2021, 2022] – a different measure than ours. Moreover, there is no clear way to translate any of these results into tabular problems.

The special case of one-step lookahead, where immediate rewards are observed before making a decision, has been studied in various problems. Possibly the most famous instance of such a problem is the prophet inequality. There, a set of known distributions is sequentially observed, and agents choose whether to either take a reward and end the interaction or discard it and move to the next distribution [Correa et al., 2019b]. This could be formulated as a chain environment with two actions – a rewarding action that moves to an absorbing state and a non-rewarding one that moves forward in the chain. A generalization of the prophet problem to resource allocation over Markov chains was studied in [Jia et al., 2023]. To obtain a CR that is independent of the interaction length, the authors allow both the online and offline algorithms to choose their initial state. In both cases (and many other problems), the CR is measured between a one-step lookahead and a full lookahead agent, which observes all rewards in advance. In contrast, we measure the CR between no-lookahead agents and all possible lookaheads, so our results are complementary.

Finally, Garg et al. [2013] studied another related resource allocation model. In their work, the competitive ratio for Markov Decision Processes is measured between an online agent with access to the $L$-future reward distributions and transition probabilities, versus an agent who observes all statistical information in advance. A similar adversarial notion is also presented specifically for resource allocation. In contrast, we assume that the distributions are known to both agents and only the oracle observes reward realizations.

## 2 Preliminaries

We work under the episodic tabular reinforcement learning model. The environment is modeled as a Markov Decision Process (MDP), defined by the tuple $(\mathcal{S}, \mathcal{A}, H, P, R, \mu)$, where $\mathcal{S}$ is the state space ($|\mathcal{S}| = S$), $\mathcal{A}$ is the action space ($|\mathcal{A}| = A$), $H \in \mathbb{N}$ is the horizon, $P$ is the transition kernel, $R$ is the stochastic reward and $\mu \in \Delta_S$ is the initial state distribution. At the first timestep, an initial state is generated $s_1 \sim \mu$. Then, at every timestep $h \in [H] \triangleq \{1, \ldots, H\}$, given environment state $s_h \in \mathcal{S}$, the agent performs an action $a_h \in \mathcal{A}$, obtains a stochastic reward $R_h(s_h, a_h)$ and transitions to a state $s_{h+1} \in \mathcal{S}$ with probability $P_h(s_{h+1}|s_h, a_h)$. For brevity, we use the notation $\mathcal{X} = [H] \times \mathcal{S} \times \mathcal{A}$.

We assume that rewards at different timesteps are independent, but allow them to be arbitrarily correlated between state-actions at the same step. We denote the expected reward by $r_h(s, a)$ and assume that the rewards are non-negative.[1] Rewards and transitions are always assumed to be mutually independent, and transitions are independent between rounds. While we focus on non-stationary models, where the reward and transition distributions could depend on the timestep $h$, our analysis techniques could be easily adapted to stationary models, where the distributions are timestep-independent, and all the proofs in the appendix also state the results for stationary models.

### 2.1 Lookahead Policies and Values

We assume w.l.o.g. that all rewards are generated before the interaction starts. We denote by $\mathcal{R}_h = \{R_h(s, a)\}_{s \in \mathcal{S}, a \in \mathcal{A}}$, the set of all rewards at timestep $h$ and by $\mathcal{R}_h^L = \{\mathcal{R}_t\}_{t=h}^{h+L-1}$, the $L$-lookahead reward information, containing all reward information for $L$-timesteps starting from $h$. By convention, $\mathcal{R}_h^0$ is the empty set. A lookahead policy is defined as follows.

**Definition 1.** *A lookahead policy $\pi^L : [H] \times \mathcal{S} \times \mathbb{R}^{SAL} \mapsto \Delta_{\mathcal{A}}$ is a policy that for each timestep $h$, observes the state $s_h$ and the lookahead reward information $\mathcal{R}_h^L$ and generates an action $a_h$ with probability $\pi_h^L(a_h|s_h, \mathcal{R}_h^L)$. The set of all lookahead policies is denoted by $\Pi^L$.*

For example, a one-step lookahead policy observes the immediate rewards at the current state before acting, while a full lookahead policy has access to all reward realizations before the interaction starts. When $L = 0$, the policy only depends on the state and is Markovian; we therefore denote $\Pi^{\mathcal{M}} = \Pi^0$.

The goal of any agent is to maximize its cumulative reward, also known as the *value*, $V^{L,\pi} = \mathbb{E}\left[\sum_{h=1}^H R_h(s_h, a_h)|s_1 \sim \mu, \pi\right]$. For brevity, we omit the conditioning on the initial state distribution. The optimal value given a lookahead $L$ is $V^{L,*} = \sup_{\pi^L \in \Pi^L} V^{L,\pi^L}$. If we want to emphasize that an environment parameter (say, the transition kernel $P$) is fixed, we shall specify it, e.g., $V^{L,\pi}(P, r)$.

We analyze the relation between the 'standard value' of an agent that plays optimally using no future information ($V^{0,*}$) and a lookahead agent that observes the $L$-future rewards before acting ($V^{L,*}$). Formally, let $\mathcal{D}(r)$ be the set of all non-negative distributions with rewards expectations $r_h(s, a)$. The $L$-lookahead competitive ratio (CR) is defined as

$$CR^L(P, r) = \inf_{\mathcal{R}^H \sim \mathcal{D}(r)} \frac{V^{0,*}(P, r)}{V^{L,*}(P, r)}. \tag{1}$$

That is, the competitive ratio is the worst-possible multiplicative loss of the standard (no-lookahead) policy, compared to an $L$-lookahead policy, given fixed transition kernel and expected rewards. For ratios to be well-defined, we follow the convention that any division by zero equals $+\infty$.

**Remark 1.** *We emphasize that the reward distributions are known in advance to both the no-lookahead and the $L$-step lookahead agents, in striking contrast to adversarial settings. In the latter, the reward could be arbitrary and is only given to an oracle agent. In particular, any upper bound on $CR^L(P, r)$ will also apply to adversarial settings.*

**Remark 2.** *Without lookahead information, $P$ and $r$ suffice to calculate the optimal value [Sutton and Barto, 2018], so one could also write $CR^L(P, r) = \frac{V^{0,*}(P,r)}{\sup_{\mathcal{R}^H \sim \mathcal{D}(r)} V^{L,*}(P,r)}$.*

We similarly study the $L$-lookahead CR for the worst-case reward expectations, defined as[2] $CR^L(P) = \inf_{r_h \in [0,1]^{SA}} CR^L(P, r)$. Finally, we study the CR for the worst-case environment

---

[1] This assumption is standard when performance is measured by ratios – otherwise, ratios are not well-defined.
[2] While we limit the expectations to $[0, 1]$, the same results hold for $r_h \in \mathbb{R}_+^{SA}$ (see Remark 3 in the appendix).

$P$ and initial state distribution $\mu$, denoted by $CR^L$. In particular, we show that stationary environments achieve near-worst-case CR.

**Interpretation: the gain from lookahead information.** The no-lookahead agent is the standard agent used throughout the RL literature and serves as an 'off-the-shelf' agent. As such, the competitive ratios quantify the potential gain when moving from classic RL settings to agents that utilize future reward information. While using future information always increases the value, it often comes at some price – either because access to such information is costly, or since lookahead algorithms are much more complicated and computationally expensive. The CRs analyzed in the paper can help determine whether the potential gain is worth the price – and choosing which agent to deploy.

## 2.2 Occupancy Measures

Occupancy measures are the visitation probabilities of an agent in different state-actions. In particular, for any (potentially lookahead) policy, we define $d_h^\pi(s) = \Pr\{s_h = s\}$ and $d_h^\pi(s,a) = \Pr\{s_h = s, a_h = a\}$, where randomness is w.r.t. both transitions, rewards and internal policy randomization, given that actions are generated from the policy $\pi \in \Pi^L$. For $h = 1$, the state distribution only depends on the initial state distribution $\mu$, and we use $d_1^\pi(s)$, $d_1(s)$ and $\mu(s)$ interchangeably. We also define the conditional occupancy measure as $d_h^\pi(s|s_t = s') = \Pr\{s_h = s|s_t = s'\}$ for some $t \leq h$ and similarly use $d_h^\pi(s,a|s_t = s')$. Intuitively, this is the reaching probability from state $s'$ at time $t$ to a state $s$ at time $h$ when playing a policy $\pi$. Without lookahead information, it is well-known that the set of occupancy measures induced by Markovian policies is a convex compact polytope [Altman, 2021], and the value of any Markovian policy could be expressed using occupancies by

$$V^{0,\pi} = \mathbb{E}\left[\sum_{h=1}^{H} R_h(s_h, a_h)\right] = \mathbb{E}\left[\sum_{(h,s,a)\in\mathcal{X}} \mathbb{1}\{s_h = s, a_h = a\} R_h(s,a)\right]$$

$$= \sum_{(h,s,a)\in\mathcal{X}} \Pr\{s_h = s, a_h = a\}\mathbb{E}[R_h(s,a)] = \sum_{(h,s,a)\in\mathcal{X}} d_h^\pi(s,a) r_h(s,a) = d^{\pi T} r. \quad (2)$$

Finally, denote the optimal reaching probability to a state $s \in \mathcal{S}$ as $d_h^*(s) = \max_{\pi\in\Pi^\mathcal{M}} d_h^\pi(s)$. Notice that rewards and transitions are independent, so reward information does not affect the optimal reaching probability and it is sufficient to look at Markovian policies. Moreover, after reaching a state $s$, an agent could always deterministically choose an action $a$, so $d_h^*(s,a) = \max_{\pi\in\Pi^\mathcal{M}} d_h^\pi(s,a) = d_h^*(s)$. Similarly, we define the optimal conditional reaching probability as $d_h^*(s|s_t = s') = \max_{\pi\in\Pi^\mathcal{M}} d_h^\pi(s|s_t = s')$, and as the for non-conditional occupancy measures, we have that $d_h^*(s,a|s_t = s') = d_h^*(s|s_t = s')$.

## 3 Competitiveness Versus Full Lookahead Agents

Before analyzing the CR for the full range of lookahead values, we start by studying the full lookahead case, where all rewards are observed before the interaction starts. This regime is applicable, for example, in goal-oriented problems, where goals are given to the agent before an episode starts [Andrychowicz et al., 2017]. Notably, we show a link between the CR for the worst-case reward expectations, $CR^H(P)$, and existing complexity measures in offline RL and reward-free exploration. While the results of this section will later be covered by the more general multi-step lookahead, this case gives valuable insights on the worst-case distributions. Moreover, much of the proof techniques presented in this section will later be used to prove the results for the multi-step lookahead.

When all rewards are observed before the interaction starts, each instantiation of the reward is equivalent to an RL problem with *known deterministic* rewards. In particular, the optimal policy given the reward is Markovian, and using the value formulation in Equation (2), we have

$$V^{H,*}(P,r) = \mathbb{E}\left[\max_{\pi\in\Pi^\mathcal{M}} \sum_{(h,s,a)\in\mathcal{X}} d_h^\pi(s,a) R_h(s,a)\right] \leq \mathbb{E}\left[\sum_{(h,s,a)\in\mathcal{X}} \max_{\pi\in\Pi^\mathcal{M}} d_h^\pi(s,a) R_h(s,a)\right]$$

$$= \sum_{(h,s,a)\in\mathcal{X}} d_h^*(s)\mathbb{E}[R_h(s,a)] = \sum_{(h,s,a)\in\mathcal{X}} d_h^*(s) r_h(s,a). \quad (3)$$

At first glance, this bound seems extremely crude – the agent optimally navigates to collect all the expected rewards. Yet, at a second glance, it gives intuition on the worst-case distribution: a situation where only one reward at a single state is realized in every episode. Then, full lookahead agents can optimally navigate to this state and still collect all the realized rewards. While we cannot fully enforce a single reward realization (due to the independence of rewards in different timesteps), we can approximate this behavior by focusing on *long-shot* distributions [Hill and Kertz, 1981].

**Definition 2.** *Rewards have long-shot distributions with parameter $\epsilon \in (0,1)$ and expectation $r$ if*

$$\forall h \in [H], s \in \mathcal{S}, a \in \mathcal{A}: \qquad R_h(s,a) = \left\{ \begin{array}{ll} r_h(s,a)/\epsilon & w.p. \ \epsilon \\ 0 & w.p. \ 1-\epsilon \end{array} \right.$$

*independently for all $h, s, a$. We also use the notation $R \sim LS_\epsilon(r)$.*

Notice that for any given $\epsilon$, long-shot distributions are bounded; thus, long-shot rewards could always be scaled to be supported by $[0,1]$ without affecting the CR. Moreover, when $\epsilon \ll 1/SAH$, with high probability, at most a single reward will be realized, and the bound in Equation (3) is achieved in equality as $\epsilon \to 0$. Formally, the CR versus a full lookahead agent is characterized as follows:

**Theorem 1.** *[CR versus Full Lookahead Agents; see Appendix A for the proof]*

*Worst-case distributions:* $CR^H(P,r) = \max_{\pi \in \Pi^{\mathcal{M}}} \frac{\sum_{(h,s,a) \in \mathcal{X}} d_h^\pi(s,a) r_h(s,a)}{\sum_{(h,s,a) \in \mathcal{X}} d_h^*(s) r_h(s,a)}.$

*Worst-case reward expectations:* $CR^H(P) = \max_{\pi \in \Pi^{\mathcal{M}}} \min_{(h,s,a) \in \mathcal{X}} \frac{d_h^\pi(s,a)}{d_h^*(s)}.$

*Worst-case environments:* *For all environments, $CR^H \geq \max\{\frac{1}{SAH}, \frac{1}{A^H}\}$. Also, for any $\delta \in (0,1)$ there exist stationary environments with rewards over $[0,1]$ s.t. if $S = A^n + 1$ for $n \in \{0, \ldots, H-1\}$, then $CR^H(P,r) \leq \frac{1+\delta}{(H - \log_A(S-1)) \cdot (A-1)(S-1)}$, and if $S \geq A^H - 1$, then $CR^H(P,r) \leq \frac{1+\delta}{A^H}$.*

*Proof Sketch.* **Part I.** Recalling Remark 2 and Equation (2), to prove the first part of the proposition, one only needs to calculate the full lookahead value for the worst-case distribution. An upper bound for this value is already given in Equation (3); we directly calculate the value for long-shot distributions $LS_\epsilon(r)$ and show that this bound is achieved at the limit of $\epsilon \to 0$.

**Part II.** The proof of the second part of the theorem utilizes the previously calculated $CR^H(P,r)$ to optimize for the worst-case expectations. This is done using the minimax theorem, exchanging the reward minimization and the policy maximization. To make the internal maximization problem concave, we move from the space of Markovian policies to the set of occupancy measures induced by Markovian policies, which is convex [Altman, 2021]. To make the reward minimization convex, we show that the denominator can be converted to the constraint $\sum_{(h,s,a) \in \mathcal{X}} d_h^*(s) r_h(s,a) = 1$. Then, the minimax theorem can be applied, and we explicitly solve the resulting optimization problem. The formal application of the minimax theorem and its solution is done in Lemma 1 in the appendix.

**Part III.** The proof of the final statement is further divided into two parts.

*Lower bounding $CR^H$.* The lower bound $CR^H \geq 1/A^H$ is inductively achieved from the dynamic programming equations for both the no-lookahead and full lookahead values. The bound $CR^H \geq 1/SAH$ is obtained by choosing a specific policy $\pi \in \Pi^{\mathcal{M}}$ and substituting in $CR^H(P)$: the Markovian policy whose occupancy is $d_h^{\pi_u}(s,a) = \frac{1}{SAH} \sum_{(h',s',a') \in \mathcal{X}} d_h^{\pi_{h',s',a'}^*}(s,a)$, where $\pi_{h,s,a}^* \in \Pi^{\mathcal{M}}$ is a policy that maximizes the reaching probability to $(h,s,a) \in \mathcal{X}$.

*Upper bounding $CR^H$ – designing a worst-case environment.* We show that a modified tree graph achieves a near-worst-case competitive ratio. In tree-based MDPs, each state represents a node in a tree, with the initial state as its root, and actions take the agent downwards through the tree. In our example, rewards are long-shots located at the leaves of such trees. However, this structure, by itself, does not lead to the worst-case bound. Intuitively, a standard RL agent would navigate to the leaf with the maximal expected reward, while an agent with a full lookahead would navigate to the leaf with the highest reward realization. Since there are at most $S$ leaves with $A$ actions in each, this would lead to $CR^H(P) \approx \frac{1}{SA}$. This is improved by a simple modification: at the root of the tree, we allocate one action to 'delay' the entrance to the tree and stay in the root (as illustrated in Figure 2 in the appendix). While agents without lookahead have no incentive to use this action, a full lookahead agent could predict *when* a reward will be realized and enter the tree at a timing that allows

its collection. When $H$ is large enough (compared to the tree depth), this allows the full lookahead agent to have approximately $H$ attempts to collect a reward and lead to the additional $H$-factor (up to log factors). The proof could be extended to any value of $S$ by allowing the tree to be incomplete – we refer the readers to the remark at the end of Proposition 1 in the appendix for more details. $\square$

Surprisingly, the CR for the worst-case reward expectation $CR^H(P)$ is the inverse of a concentrability coefficient that appears in many different RL settings, called the *coverability coefficient*. In particular, it affects the learning complexity in both online and offline RL settings, where agents must learn to act optimally either based on logged date or interaction with the environment [Xie et al., 2022].[3] It also has a central role in reward-free exploration, where agents aim to learn the environment so that they can perform well for *any* given reward function [Al-Marjani et al., 2023]. We emphasize that the lookahead setting is fundamentally different – we assume that all agents have exact information on both the dynamics and reward distributions and ask about the multiplicative performance improvement due to additional knowledge on reward realization. In contrast, in learning settings, the main complexity is usually in learning the dynamics, and the rewards are oftentimes assumed to be deterministic. Moreover, the analyzed quantities are either regret measures or sample complexity, which cannot be directly linked to the competitive ratio.

The last part of Theorem 1 shows that tree-like environments with a delaying action at their root exhibit worst-case CR. Similar delay mechanisms were previously used to prove regret and PAC lower bounds for nonstationary MDPs [Domingues et al., 2021, Tirinzoni et al., 2022], though with a major difference – in previous works, a nonstationary reward distribution is used to force the agent to learn when to traverse the tree and where to navigate, and the reward is time-extended (obtained for $\Omega(H)$ rounds). In contrast, our formulation is fully stationary and a reward can only be collected once. Still, the lookahead agent can use the delay to linearly increase the reward-collection probability, without any need to create time-extended rewards.

## 4 Competitiveness Versus Multi-Step Lookahead Agents

We now generalize the results of Section 3 and analyze the competitive ratio compared to $L$-lookahead agents, for *any* possible lookahead range $L \in [H]$. We also give special attention to the case of one-step lookahead, where the immediate rewards are revealed *before* taking an action.

Inspired by the full lookahead case, we focus on long-shot rewards. For such rewards, an agent would expect to see no more than a single reward during an episode, which would only be discovered $L$-steps in advance. As such, a reasonable strategy would play a Markovian policy that maintains a 'favorable' state distribution, such that whenever and wherever a future reward is realized, the agent could optimally navigate to it. Letting $t_L(h)$ be the time step where the $h$-step rewards are revealed to an $L$-lookahead agent, this corresponds with the following worst-case value:

**Proposition 2.** *For any $L \in [H]$, let $t_L(h) = \max\{h - L + 1, 1\}$. Then, it holds that*

$$\sup_{\mathcal{R}^H \sim \mathcal{D}(r)} V^{L,*}(P, r) = \max_{\pi \in \Pi^{\mathcal{M}}} \sum_{(h,s,a) \in \mathcal{X}} r_h(s,a) \sum_{s' \in \mathcal{S}} d^{\pi}_{t_L(h)}(s') d^*_h(s | s_{t_L(h)} = s')$$

The proof can be found at Appendix B. It is comprised of calculating the value of long-shot rewards $R \sim LS_\epsilon(r)$ at the limit when $\epsilon \to 0$ and then showing that the same quantity also serves as an upper bound of the value for all reward distributions.

For full lookahead, we have $t_H(h) = 1$, and $d^{\pi}_{t_H(h)}$ becomes the initial state distribution $\mu$. This leads to the same value as in Equation (3). The second extremity is when $L = 1$ and $t_1(h) = h$. Then, the conditional occupancy is $d^*_h(s | s_{t_L(h)} = s') = \mathbb{1}\{s = s'\}$ and we get the simplified expression

$$\sup_{\mathcal{R}^H \sim \mathcal{D}(r)} V^{1,*}(P, r) = \max_{\pi \in \Pi^{\mathcal{M}}} \sum_{(h,s,a) \in \mathcal{X}} r_h(s,a) d^{\pi}_h(s). \tag{4}$$

Notably, this is the value of an agent that collects the rewards of all the actions in visited states (regardless of the action it actually played) but has no lookahead information.

---

[3] A subtle difference between the coefficients is whether the outer maximum is over all valid occupancy measures or all possible state-action distributions; see [Al-Marjani et al., 2023, Section 2.3] for further discussion.

Recalling Remark 2, one could use Proposition 2 to directly calculate $CR^L(P, r)$. This, in turn, allows analyzing the worst-case reward expectations and environment, as stated in the following:

**Theorem 3.** *[CR versus Multi-Step Lookahead Agents; see Appendix B for the proof]*

*For any $L \in [H]$, let $t_L(h) = \max\{h - L + 1, 1\}$. Then, it holds that:*

*Worst-case distributions:* $CR^L(P, r) = \dfrac{\max_{\pi \in \Pi^{\mathcal{M}}} \sum_{(h,s,a) \in \mathcal{X}} r_h(s,a) d_h^\pi(s,a)}{\max_{\pi \in \Pi^{\mathcal{M}}} \sum_{(h,s,a) \in \mathcal{X}} r_h(s,a) \sum_{s' \in \mathcal{S}} d_{t_L(h)}^\pi(s') d_h^*(s|s_{t_L(h)} = s')}.$

*Worst-case reward expectations:*

$$CR^L(P) = \min_{\pi^* \in \Pi^{\mathcal{M}}} \max_{\pi \in \Pi^{\mathcal{M}}} \min_{(h,s,a) \in \mathcal{X}} \frac{d_h^\pi(s,a)}{\sum_{s' \in \mathcal{S}} d_{t_L(h)}^{\pi^*}(s') d_h^*(s|s_{t_L(h)} = s')}.$$

*Worst-case environments:* *For all environments, $CR^L \geq \max\left\{\frac{1}{SAH}, \frac{1}{(H-L+1)A^L}\right\}$. Also, for any $\delta \in (0,1)$ there exist stationary environments with rewards over $[0,1]$ s.t. if $S = A^n + 1$ for $n \in \{0, \ldots, L-1\}$, then $CR^L(P, r) \leq \frac{1+\delta}{(H-\log_A(S-1))\cdot(A-1)(S-1)}$, and if $S \geq A^L + 1$, then $CR^L(P, r) \leq \frac{1+\delta}{(H-L+1)(A^L-1)}$.*

*Proof Sketch.* The first part of the theorem is a direct result of Proposition 2 and Remark 2. For the second part, we first rewrite

$$CR^L(P, r) = \min_{\pi^* \in \Pi^{\mathcal{M}}} \max_{\pi \in \Pi^{\mathcal{M}}} \frac{\sum_{(h,s,a) \in \mathcal{X}} r_h(s,a) d_h^\pi(s,a)}{\sum_{(h,s,a) \in \mathcal{X}} r_h(s,a) \sum_{s' \in \mathcal{S}} d_{t_L(h)}^{\pi^*}(s') d_h^*(s|s_{t_L(h)} = s')},$$

and as in the full lookahead case, we apply the minimax theorem using Lemma 1. However, direct application would require calculating the infimum over $\pi^* \in \Pi^{\mathcal{M}}$, and not a minimum. Thus, compared to the full lookahead, we also need to prove that the minimum is obtained in this set. We do so in Lemma 2, relying on the set of occupancy measures being a convex compact polytope.

In the last part, we use the same tree example to upper bound $CR^L(P, r)$. The lower bound is proven using a reduction from the full lookahead bound. In particular, the bound of $1/SAH$ trivially holds from the full lookahead case. For the second lower bound, we devise a Markovian policy $\pi_u$ such that for the appropriate choice of reward functions $r^i$, we prove that

$$\frac{V^{0,\pi_u}(P, r)}{V^{L,*}(P, r)} \geq \frac{1}{H-L+1} \min_{\substack{i \in [H-L+1], \\ s' \in \mathcal{S}}} \left\{ \frac{\max_{\pi \in \Pi^{\mathcal{M}}} \sum_{(s,a) \in \mathcal{S} \times \mathcal{A}} \sum_{h=i}^{i+L-1} r_h^i(s,a) d_h^\pi(s,a|s_i = s')}{\sum_{(s,a) \in \mathcal{S} \times \mathcal{A}} \sum_{h=i}^{i+L-1} r_h^i(s,a) d_h^*(s|s_i = s')} \right\}.$$

Each of the terms is the competitive ratio versus a full lookahead agent with horizon $L$ that starts acting at $s_i = s'$. Hence, by Theorem 1, all terms are bounded by $\frac{1}{A^L}$. To elaborate, the reward $r^i$ limits the reward only to the new timesteps the lookahead agent gets to observe when it reaches step $i$. The policy $\pi_u$ is a mixture (in the occupancy space) of policies $\pi_i$ that start by playing the Markovian policy that maximizes the value of Proposition 2, up to timestep $i$, and then maximizes $r^i$. $\square$

Theorem 3 extends the full lookahead results of Theorem 1 and tightly characterizes the CR for the full spectrum of lookaheads, both as a function of the environment and for the worst-case environments. Notice that even though lookahead policies are highly non-Markovian, all bounds are expressed using Markovian policies.

**One-step lookahead.** In the case where the immediate reward is observed before acting, Theorem 3 proves that even for the worst-case environment, $CR^1 = \Theta\left(\frac{1}{HA}\right)$, namely, independent of the size of the state-space. Moreover, for any transition kernel $P$, the CR is given by

$$CR^1(P) = \min_{\pi^* \in \Pi^{\mathcal{M}}} \max_{\pi \in \Pi^{\mathcal{M}}} \min_{(h,s,a) \in \mathcal{X}} \frac{d_h^\pi(s,a)}{d_h^{\pi^*}(s)}. \tag{5}$$

While the coverability coefficient of $CR^H(P)$ requires a policy $\pi$ to cover all states *simultaneously* in proportion to their optimal reaching probability, $CR^1(P)$ provides a weaker coverability notion; it requires being able to cover *any pre-known* state-distribution induced by a Markov policy $\pi^*$. We emphasize that $\pi$ must cover this distribution using all actions, so imitating the behavior of $\pi^*$ might be challenging – with a ratio of $1/AH$ as the worst case.

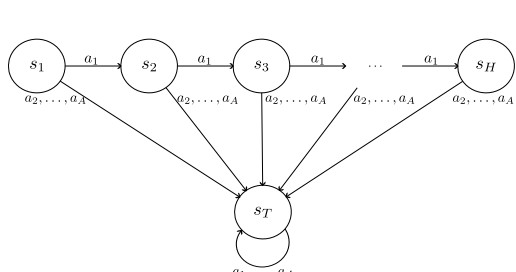
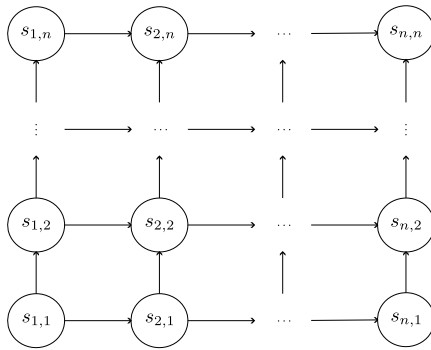

(a) Chain MDP: agents start at the head of a chain and can either move forward in the chain or transition to an absorbing terminal state.

(b) Grid MDP: agents start at the bottom-left corner of an $n \times n$ grid and can move either up or right, until ending at the top-right corner after $2n - 1$ steps.

Figure 1: Examples: CR for grid and chain environments.

Thus, $CR^1(P)$ could be seen as an intermediate point between the coverability coefficient and *single-policy coverability* [Xie et al., 2022], defined by the ratio between the state-action occupancy of the optimal policy and a single data distribution. Yet, Xie et al. [2022] argue that this notion is too weak to allow any guarantees. It is of interest to investigate whether our refined notion, which requires covering all valid state distributions, mitigates the issues they present and allows deriving meaningful results in offline and online RL.

In general, one could interpret the ratios $CR^L(P)$ as a class of decreasing[4] (inverse) concentrability coefficients, starting from the coverability of all pre-known state distributions ($CR^1(P)$) and ending with the coverability coefficient ($CR^H(P)$). Thus, it is intriguing to further study the connection of these values to other domains in which concentrability naturally arises.

## 5 Examples

We now present several MDP structures and analyze their competitive ratio for various lookaheads.

**Disguised contextual bandit [Al-Marjani et al., 2023].** Maybe the most basic scenario is when actions do not affect the transitions, i.e., $P_h(s'|s,a) = P_h(s'|s)$ for all possible $(h,s,a,s')$. Specifically, the state distribution is independent of the played policy – there exists an occupancy measure $d_h$ such that for all policies, $d_h^\pi(s) = d_h(s)$. Thus, it also holds that $d_h^*(s) = d_h(s)$, and

$$CR^H(P) = \max_{\pi \in \Pi^\mathcal{M}} \min_{h,s,a} \frac{d_h^\pi(s,a)}{d_h^*(s)} = \max_{\pi \in \Pi^\mathcal{M}} \min_{h,s,a} \frac{d_h(s)\pi_h(a|s)}{d_h(s)} = \max_{\pi \in \Pi^\mathcal{M}} \min_{h,s,a} \pi_h(a|s) = \frac{1}{A}.$$

The last equality holds since $\pi_h(a|s) \in \Delta_\mathcal{A}$. Using the same arguments, one could also obtain this CR for one-step lookahead, so by the monotonicity of the CR in the lookahead, $CR^L(P) = \frac{1}{A}$ for all $L \in [H]$. This is to be expected – without control over the dynamics, the best lookahead agents could do is to maximize immediate rewards, and any additional lookahead information is useless. Then, in each state, knowing the realization can only increase the reward by a factor of $\frac{\mathbb{E}[\max_a R_h(s,a)]}{\max_a \mathbb{E}[R_h(s,a)]} \leq A$.

**Delayed trees.** This is the example described in the proofs of the main results, also detailed in Proposition 1 and depicted in Figure 2. In such environments, we get a worst-case CR of $CR^L(P, r) = \Theta\left(\max\left\{\frac{1}{(H-L)A^L}, \frac{1}{SAH}\right\}\right)$. These trees are an extreme case where lookahead information is not only used to collect immediate rewards but rather to navigate to long-term rewards.

**Chain MDPs.** We go back to a bandit-like scenario and add limited control on the dynamics, in the form of a chain. The agent starts at the head of the chain ($s_1$), and at each node $k$ of the chain, it could choose to advance to the next node by taking the action $a = a_1$ or to move to an absorbing terminal state $s_T$ by taking any other action. The environment is depicted in Figure 1(a).

---

[4]The sequence is decreasing by definition because increasing the lookahead only extends the policy class.

One special problem that falls into this structure is the prophet inequality problem. In particular, assume that reward can only be obtained when moving from the chain to the terminal state ($\forall k, a,\ r_h(s_k, a_1) = r_h(s_T, a) = 0$). Thus, at each node of the chain, the agent chooses whether to collect a reward and effectively end the interaction or discard it and move forward in the chain. In other words, the problem becomes an optimal-stopping problem. As such, it is reasonable to allow the agent to see the instantaneous rewards before deciding whether to stop, leading to one-step lookahead agents. This problem has numerous applications, especially in the context of posted-price mechanisms [Correa et al., 2017, 2019a]. A classical result is that the CR between one-step lookahead and full lookahead agents is always bounded by $1/2$ [Hill and Kertz, 1981].

Assuming this reward structure with the worst-case reward distribution, the full lookahead agent could reach all rewards and collect them, thus collecting $V^{H,*}(P, r) = \sum_{k=1}^{H} \sum_{a \in \mathcal{A}} r_k(s_k, a)$ (as in Equation (3)). Similarly, a one-step lookahead agent could move forward in the chain using the policy $\pi_k(s_k) = a_1$ while effectively collecting all rewards and achieving the same value (see Equation (4)). In contrast, a no-lookahead agent would have to choose a single reward to collect, obtaining a value of $V^{0,*}(P, r) = \max_{k \in [H], a \in \mathcal{A}} r_k(s_k, a)$. The resulting CR for this reward structure would be

$$CR^H(P, r) = CR^1(P, r) = \frac{\max_{k \in [H], a \in \mathcal{A}} r_k(s_k, a)}{\sum_{k=1}^{H} \sum_{a \in \mathcal{A}} r_k(s_k, a)} \geq \frac{1}{(A-1)H},$$

where the inequality is since there are only $A - 1$ rewarding actions, and equality is achieved when all expected rewards are equal. Notably, the reward structure in the prophet problem is near-worst-case; one could verify that for chain MDPs, it holds that $CR^H(P) \geq \left(1 - \frac{1}{e}\right)\frac{1}{AH}$. This is due to the second part of Theorem 1, using the following policy: for all chain states $k \in [H]$, move forward w.p. $\pi_k(a_1|s_k) = 1 - \frac{1}{H}$ and play any other action $i > 1$ w.p. $\pi_k(a_i|s_k) = \frac{1}{(A-1)H}$. At the absorbing state $s_T$, play uniformly $\pi_h(a_i|s_T) = 1/A$. This simple example provides two important insights.

*Hardness versus one-step lookahead*: chain MDPs exhibit the worst-case CR versus one-step lookahead agents. A central reason is that to move towards rewarding states (forward in the chain), agents must take non-rewarding actions ($a_1$) – there is a tradeoff between gathering instantaneous rewards and moving to future rewarding states.

*Easiness versus full lookahead*: as previously mentioned, the CR between one-step and full lookahead agents is the well-known prophet inequality and is at least $1/2$; In other words, for chain MDPs, the information-gain from one-step-to full lookahead is marginal compared to the value of one-step versus no-lookahead. This is mainly because navigating to rewarding states is especially easy in chain MDPs – the agent only has to move forward. In contrast, in environments where navigating to rewarding states is difficult (e.g., the tree environment described in the main results), there is a substantial gain to the full lookahead.

These insights motivate two natural assumptions that reduce the CR.

**Dense rewards.** Assume that in all states where the reward can be strictly positive, it holds that $\frac{\max_a r_h(s,a)}{\min_a r_h(s,a)} \leq C$. That is, if there exists one rewarding action at a state, all its actions yield some minimal reward. States are allowed to yield zero rewards for all actions. When this assumption holds, agents could navigate to rewarding future states and still collect rewards, mitigating the issue observed in the chain MDPs. Letting $\pi^* \in \arg\max_{\pi \in \Pi^{\mathcal{M}}} \sum_{(h,s,a) \in \mathcal{X}} r_h(s, a)d_h^\pi(s)$, we have

$$CR^1(P, r) = \frac{\max_{\pi \in \Pi^{\mathcal{M}}} \sum_{(h,s,a) \in \mathcal{X}} r_h(s,a)d_h^\pi(s,a)}{\max_{\pi \in \Pi^{\mathcal{M}}} \sum_{(h,s,a) \in \mathcal{X}} r_h(s,a)d_h^\pi(s)} \geq \frac{\sum_{(h,s,a) \in \mathcal{X}} r_h(s,a)d_h^{\pi^*}(s,a)}{\sum_{(h,s,a) \in \mathcal{X}} r_h(s,a)d_h^{\pi^*}(s)}$$

$$\geq \frac{\sum_{(h,s,a) \in \mathcal{X}} \frac{1}{AC} \sum_{a' \in \mathcal{A}} r_h(s,a')d_h^{\pi^*}(s,a)}{\sum_{(h,s,a) \in \mathcal{X}} r_h(s,a)d_h^{\pi^*}(s)} \overset{(*)}{\geq} \frac{1}{AC},$$

where $(*)$ is since $\sum_a d_h^{\pi^*}(s,a) = d_h^{\pi^*}(s)$. Thus, dense rewards remove the horizon dependence in the CR, and for small $C$, we get a similar CR as in the disguised contextual bandit problem.

**Ergodic MDPs.** One way to make the navigation task easier is to limit the control of the agent on the state. In [Al-Marjani et al., 2023], the authors suggest looking at MDPs whose transition kernels are near-uniform. Formally, for $0 < \beta < \alpha < 1$, they defined the family of transitions

$$\mathcal{P}_{\alpha,\beta} = \left\{ q \in \mathbb{R}_+^S : \sum_{i=1}^S q_i = 1, \max_i q_i \leq S^{\alpha-1}, \min_i q_i \geq \frac{1 - S^{\beta-1}}{S-1} \right\},$$

and assumed that $P_h(\cdot|s,a) \in \mathcal{P}_{\alpha,\beta}$ for all $h,s,a$. As $\alpha$ goes to zero, the transition distribution becomes uniform, while at the limit of $\alpha, \beta \to 1$, this becomes the set of all possible transition kernels. Under this assumption, they prove that the coverability coefficient is bounded by $S^\alpha AH$ (see the end of the proof of Lemma 38 of Al-Marjani et al. 2023), which implies that $CR^H(P) \geq \frac{1}{S^\alpha AH}$. In particular, if for all $h,s,a$, $P_h(s'|s,a) \in \left[ \frac{1-C/S}{S-1}, \frac{C}{S} \right]$, then $CR^H(P) \geq \frac{1}{CAH}$: independent of the size of the state-space. Finally, in their proof, Al-Marjani et al. 2023 show that $d_h^\pi(\cdot) \in \mathcal{P}_{\alpha,\beta}$ for all policies and timesteps. Substituting to Equation (5) (and using the uniform policy for $\pi$) directly leads to $CR^1(P) \geq \frac{1-S^{\beta-1}}{AS^\alpha}$, potentially improving the worst-case environment when $S^\alpha \leq H$.

**Grid MDPs** We end this section by analyzing a navigation example, where an agent navigates from one corner of an $n \times n$ grid to the opposite corner ("Navigating in Manhattan", see Figure 1(b)). Due to space limits, we briefly describe the results while fully proving them in Appendix C.2. This example directly generalizes the chain example with added navigation difficulty; by enforcing zero rewards for all states above the bottom row, we effectively get a chain MDP of horizon $n$. As a direct result, we immediately get that $CR^1(P) = \Theta(\frac{1}{H})$ and $CR^H(P) = \mathcal{O}(\frac{1}{H})$. Surprisingly, this bound is tight – adding one additional dimension to the problem is just as difficult as a chain. Like chains, some of the difficulty comes from sparsity in the reward, but even when all rewards have unit expectations, we show that $CR^L(P) = \Theta(\frac{1}{L})$. This implies that the problem has additional hardness due to the need for navigation, which is the same order of magnitude as the one due to sparse reward. As a final remark, we show that the ratio between one-step lookahead and full lookahead in grid MDPs is at most $\mathcal{O}(\frac{1}{H})$. This might be counter-intuitive at first, as the worst-case CR versus either of them is $\Theta(\frac{1}{H})$. In fact, this is possible since the worst-case environments are different; when competing with one-step lookahead agent, the hardness comes from reward sparsity, while versus full lookahead, it is also due to navigation issues. The one-step lookahead agent cannot use its information to navigate, so it has the same CR of $1/H$ as the no-lookahead.

## 6 Conclusions and Future Work

We studied the value of future reward lookahead information in tabular reinforcement learning through the lens of competitive analysis. We characterized the CR for the worst-case distributions, reward expectations and transition kernels for the full range of possible lookahead. We also showed the connection between the resulting CR and concentrability coefficients from the literature of offline and reward-free RL. We find the appearance of the same coefficients in seemingly completely different RL problems intriguing and warrants further study.

While we took the first step in analyzing competitiveness in RL, various other competitive measures could be studied. One natural alternative would be to study transition lookahead, where agents observe future transition realizations. We believe that the results would greatly differ from ours; indeed, even with one-step lookahead, the CR can be exponentially small (as we prove in Appendix C.3). Another relevant competitivity measure is to compare an agent with *predictions* of the future rewards to agents with exact lookahead information. This models the realistic scenario where agents get approximate information on future rewards and want to utilize it to improve performance. Also, as in the prophet problem, one could analyze the CR between multi-step lookahead to full lookahead agents.

Finally, we focus on the CR for the worst-case distribution, which allows us to derive the exact value of lookahead agents. However, planning with lookahead for general reward distribution can be challenging. For full lookahead, one can perform standard planning using reward realization, making planning tractable. With one-step lookahead, it is possible to write Bellman equations for the value, but each calculation depends on the full distribution of the reward, making it hard to calculate. For multi-step lookahead, there is no clear way to perform planning without incorporating the future rewards into the state, rendering the planning exponential. While exact planning might be intractable, it could be possible to devise methods for approximate planning. Lastly, it is of great interest to design practical algorithms that can efficiently leverage lookahead information, that is, achieve the lookahead value; our results indicate that it is significantly higher than the no-lookahead value, so aiming for it could dramatically boost the performance. We also leave this direction for future work.

## Acknowledgments

We thank Simon Mauras and Jose Correa for the helpful discussions. This project has received funding from the European Union's Horizon 2020 research and innovation programme under the Marie Skłodowska-Curie grant agreement No 101034255. Dorian Baudry thanks the support of ANR-19-CHIA-02 SCAI.

Vianney Perchet's research was supported in part by the French National Research Agency (ANR) in the framework of the PEPR IA FOUNDRY project (ANR-23-PEIA-0003) and through the grant DOOM ANR-23-CE23-0002. It was also funded by the European Union (ERC, Ocean, 101071601). Views and opinions expressed are however those of the author(s) only and do not necessarily reflect those of the European Union or the European Research Council Executive Agency. Neither the European Union nor the granting authority can be held responsible for them.

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

# A  Proofs for Full Lookahead Agents

**Theorem 4** (CR versus Full Lookahead Agents).

*Worst-case distributions:* $CR^H(P, r) = \max_{\pi \in \Pi^{\mathcal{M}}} \frac{\sum_{(h,s,a) \in \mathcal{X}} d_h^\pi(s,a) r_h(s,a)}{\sum_{(h,s,a) \in \mathcal{X}} d_h^*(s) r_h(s,a)}.$

*Worst-case reward expectations:* *For non-stationary reward expectations,*

$$CR^H(P) = \max_{\pi \in \Pi^{\mathcal{M}}} \min_{(h,s,a) \in \mathcal{X}} \frac{d_h^\pi(s,a)}{d_h^*(s)}.$$

*If the reward expectations are stationary ($r_h(s,a) = r(s,a)$), then*

$$CR^H(P) = \max_{\pi \in \Pi^{\mathcal{M}}} \min_{(s,a) \in \mathcal{S} \times \mathcal{A}} \frac{\sum_{h=1}^H d_h^\pi(s,a)}{\sum_{h=1}^H d_h^*(s)}.$$

*Worst-case environments:* *For all environments, $CR^H \geq \max\{\frac{1}{SAH}, \frac{1}{A^H}\}$. Also, for any $\delta \in (0,1)$ there exist stationary environments with rewards over $[0,1]$ s.t. if $S = A^n + 1$ for $n \in \{0, \ldots, H-1\}$, then $CR^H(P,r) \leq \frac{1+\delta}{(H - \log_A(S-1)) \cdot (A-1)(S-1)}$, and if $S \geq A^H - 1$, then $CR^H(P,r) \leq \frac{1+\delta}{A^H}.$*

*Proof.* **Worst-case distribution:** We already saw in Equation (3) that for all reward distributions

$$V^{H,*}(P, r) \leq \sum_{(h,s,a) \in \mathcal{X}} d_h^*(s) r_h(s,a).$$

We now show that for any $\delta > 0$, there exists a distribution such that

$$V^{H,*}(P, r) \geq (1 - \delta) \sum_{(h,s,a) \in \mathcal{X}} d_h^*(s) r_h(s,a).$$

This would imply that

$$\sup_{\mathcal{R}^H \sim \mathcal{D}(r)} V^{H,*}(P, r) = \sum_{(h,s,a) \in \mathcal{X}} d_h^*(s) r_h(s,a)$$

and conclude this part of the proof (by Remark 2 and Equation (2)).

Let $\epsilon \in (0,1)$ and assume long-shot reward distribution $R \sim LS_\epsilon(r)$. For any $(h,s,a) \in \mathcal{X}$, define the event that a positive reward was realized just in $(h, s, a)$:

$$\mathbb{G}_{h,s,a} = \left\{ R_h(s,a) = \frac{r_h(s,a)}{\epsilon}, \forall (h', s', a') \neq (h, s, a) : R_{h'}(s', a') = 0 \right\}.$$

Under any of these events, the value of the optimal full lookahead agent is

$$\mathbb{E}\left[ \max_{\pi \in \Pi^{\mathcal{M}}} \sum_{(h',s',a') \in \mathcal{X}} d_h^\pi(s',a') R_{h'}(s',a') \Big| \mathbb{G}_{h,s,a} \right] = \mathbb{E}\left[ \max_{\pi \in \Pi^{\mathcal{M}}} d_h^\pi(s,a) \frac{r_h(s,a)}{\epsilon} \Big| \mathbb{G}_{h,s,a} \right] = \frac{r_h(s,a)}{\epsilon} d_h^*(s).$$

Now, notice that each of these mutually exclusive events $\mathbb{G}_{h,s,a}$ occur w.p. $\epsilon(1-\epsilon)^{SAH-1}$, and that the value is non-negative when none of them occur. Hence, for this reward distribution,

$$V^{H,*}(P, r) \geq \sum_{(h,s,a) \in \mathcal{X}} \Pr\{\mathbb{G}_{h,s,a}\} \mathbb{E}\left[ \max_{\pi \in \Pi^{\mathcal{M}}} \sum_{(h',s',a') \in \mathcal{X}} d_h^\pi(s',a') R_{h'}(s',a') \Big| \mathbb{G}_{h,s,a} \right]$$

$$= \sum_{(h,s,a) \in \mathcal{X}} \epsilon(1-\epsilon)^{SAH-1} \frac{r_h(s,a)}{\epsilon} d_h^*(s)$$

$$= (1-\epsilon)^{SAH-1} \sum_{(h,s,a) \in \mathcal{X}} d_h^*(s) r_h(s,a). \tag{6}$$

Setting $\epsilon = 1 - (1 - \delta)^{\frac{1}{SAH-1}} \approx \frac{\delta}{SAH}$ leads to the desired bound and concludes this part of the proof.

**Worst-case reward expectations:** Before proving the results, we remark on the choice to limit reward expectations to $[0,1]$. The main motivation for doing so is the ubiquity of this boundedness assumption in the literature of RL, but in fact, it is only a matter of convention and has no real impact. Indeed, since CR is invariant to scaling, the same result would directly hold for any bounded interval $[0, C]$. Furthermore, as explained in Remark 3, the result would also hold under the less restrictive assumptions that reward expectations are just non-negative.

We proof both results using Lemma 1. As a first step, we highlight that the only dependence of the optimization problem in the Markovian policy $\pi$ is through the occupancy measure $d_h^\pi$. Therefore, denoting the set of all occupancy measures induced by a Markovian policy with transition kernel $P$ by $D = D^{\mathcal{M}}(P)$, the problem can be reformulated as

$$
\begin{aligned}
CR^H(P) &= \inf_{r_h \in [0,1]^{SA}} CR^H(P, r) \\
&= \inf_{r_h \in [0,1]^{SA}} \max_{\pi \in \Pi^{\mathcal{M}}} \frac{\sum_{(h,s,a) \in \mathcal{X}} d_h^\pi(s,a) r_h(s,a)}{\sum_{(h,s,a) \in \mathcal{X}} d_h^*(s) r_h(s,a)} \\
&= \inf_{r_h \in [0,1]^{SA}} \max_{d \in D^{\mathcal{M}}(P)} \frac{\sum_{(h,s,a) \in \mathcal{X}} d_h(s,a) r_h(s,a)}{\sum_{(h,s,a) \in \mathcal{X}} d_h^*(s) r_h(s,a)}. \quad (7)
\end{aligned}
$$

The set of the possible occupancy measures is convex and compact in $\mathbb{R}_+^{SAH}$ [Altman, 2021], so we can apply Lemma 1 with $\alpha_{h,s,a} = d_h^*(s)$, $y_{h,s,a} = d_h(s,a)$ and $x_{h,s,a} = r_h(s,a)$, resulting with

$$
CR^H(P) = \max_{d \in D^{\mathcal{M}}(P)} \min_{(h,s,a) \in \mathcal{X}} \frac{d_h(s,a)}{d_h^*(s)} = \max_{\pi \in \Pi^{\mathcal{M}}} \min_{(h,s,a) \in \mathcal{X}} \frac{d_h^\pi(s,a)}{d_h^*(s)},
$$

where we again used the equivalence between optimizing over Markovian policies and their occupancy measures.

For stationary rewards, where $r_h(s,a) = r(s,a)$ for all $(h,s,a) \in \mathcal{X}$, we rewrite Equation (7) as

$$
CR^H(P) = \inf_{r_h \in [0,1]^{SA}} \max_{d \in D^{\mathcal{M}}(P)} \frac{\sum_{(s,a) \in \mathcal{S} \times \mathcal{A}} \left( \sum_{h=1}^H d_h(s,a) \right) r(s,a)}{\sum_{(s,a) \in \mathcal{S} \times \mathcal{A}} \left( \sum_{h=1}^H d_h^*(s) \right) r(s,a)}.
$$

Now, another application of Lemma 1 with $\alpha_{s,a} = \sum_{h=1}^H d_h^*(s)$, $y_{s,a} = \sum_{h=1}^H d_h(s,a)$ and $x_{s,a} = r(s,a)$ yields

$$
CR^H(P) = \max_{d \in D^{\mathcal{M}}(P)} \min_{(s,a) \in \mathcal{S} \times \mathcal{A}} \frac{\sum_{h=1}^H d_h(s,a)}{\sum_{h=1}^H d_h^*(s)} = \max_{\pi \in \Pi^{\mathcal{M}}} \min_{(s,a) \in \mathcal{S} \times \mathcal{A}} \frac{\sum_{h=1}^H d_h^\pi(s,a)}{\sum_{h=1}^H d_h^*(s)},
$$

which is the desired result for stationary environments.

**Worst-case environment – lower bound:** We now derive the lower bound $CR^H \geq \max\{\frac{1}{SAH}, \frac{1}{A^H}\}$. We prove it for nonstationary environments, so in particular, it also holds for stationary ones.

Recall that by definition, for any $(h, s, a) \in \mathcal{X}$, $d_h^*(s, a)$ is the occupancy measure of a Markovian policy that maximizes the visitation probability in $(h, s, a)$, and let $\pi_{h,s,a}^*$ be a Markovian policy that achieves this occupancy. Since the set of occupancy measures induced by Markovian policies is convex [Altman, 2021], there exists a Markovian policy $\pi_u \in \Pi^{\mathcal{M}}$ such that its occupancy measure is the average of all these occupancies, namely, for all $(h, s, a) \in \mathcal{X}$,

$$d_h^{\pi_u}(s, a) = \frac{1}{SAH} \sum_{(h', s', a') \in \mathcal{X}} d_h^{\pi_{h',s',a'}^*}(s, a).$$

Using the previous part of the theorem, for all environments $P$, it holds that

$$
\begin{aligned}
CR^H(P) &= \max_{\pi \in \Pi^{\mathcal{M}}} \min_{(h,s,a) \in \mathcal{X}} \frac{d_h^{\pi}(s, a)}{d_h^*(s)} \\
&\geq \min_{(h,s,a) \in \mathcal{X}} \frac{d_h^{\pi_u}(s, a)}{d_h^*(s)} \\
&= \min_{(h,s,a) \in \mathcal{X}} \frac{\frac{1}{SAH} \sum_{(h',s',a') \in \mathcal{X}} d_h^{\pi_{h',s',a'}^*}(s, a)}{d_h^*(s)} \\
&\overset{(*)}{\geq} \min_{(h,s,a) \in \mathcal{X}} \frac{\frac{1}{SAH} d_h^{\pi_{h,s,a}^*}(s, a)}{d_h^*(s)} \\
&= \frac{1}{SAH} \min_{(h,s,a) \in \mathcal{X}} \frac{d_h^*(s, a)}{d_h^*(s)} \\
&= \frac{1}{SAH}.
\end{aligned}
$$

In $(*)$, we discard all the (non-negative) terms in the summation where $(h', s', a') \neq (h, s, a)$, while in the following equalities, we use the definition of $\pi_{h,s,a}^*$ and the fact that $d_h^*(s, a) = d_h^*(s)$. As this inequality holds for all environments, it also implies that $CR^H \geq \frac{1}{SAH}$.

To prove that $CR^H \geq \frac{1}{A^H}$, we take a different approach and go back to the Bellman equations. Denote by $\bar{V}_h^*(s|R)$, the optimal value of a full lookahead policy, starting from timestep $h \in [H]$ and state $s \in \mathcal{S}$, and given reward realization $R$. Therefore, the value of the full lookahead agent is given by $V^{H,*} = \mathbb{E}_{R,s \sim \mu}[\bar{V}_1^*(s|R)]$. Similarly, denote the standard value with no lookahead information starting from timestep $h \in [H]$ and state $s \in \mathcal{S}$ by $V_h^{0,*}(s)$. As previously explained, given reward realizations, the optimal full lookahead policy is Markovian, so both values can be calculated using the following Bellman equations for all $s \in \mathcal{S}$ and $h \in [H]$:

$$\bar{V}_h^*(s|R) = \max_{a \in \mathcal{A}} \left\{ R_h(s, a) + \sum_{s' \in \mathcal{S}} P_h(s'|s, a) \bar{V}_{h+1}^*(s'|R) \right\}, \qquad \bar{V}_{H+1}^*(s|R) = 0$$

$$V_h^{0,*}(s) = \max_{a \in \mathcal{A}} \left\{ r_h(s, a) + \sum_{s' \in \mathcal{S}} P_h(s'|s, a) V_{h+1}^{0,*}(s') \right\}, \qquad V_{H+1}^{0,*}(s) = 0.$$

We prove by backward induction that for all $h \in [H+1]$ and $s \in \mathcal{S}$, $\mathbb{E}[\bar{V}_h^*(s|R)] \leq A^{H+1-h} V_h^{0,*}(s)$. Specifically, using this relation for $h = 1$ and taking the expectation over the initial state distribution would imply that $V^{H,*} \leq A^H V^{0,*}$, regardless of the environment, and thus $CR^H \geq \frac{1}{A^H}$.

As the base of the induction, see that the claim trivially holds for $h = H + 1$, where all values are $0$. Next, for any $h \in [H]$ and $s \in \mathcal{S}$, given that the claim holds for all states in step $h + 1$, we have

$$
\begin{aligned}
\mathbb{E}\left[\bar{V}_h^*(s|R)\right] &= \mathbb{E}\left[\max_{a \in \mathcal{A}}\left\{R_h(s,a) + \sum_{s' \in \mathcal{S}} P_h(s'|s,a)\bar{V}_{h+1}^*(s'|R)\right\}\right] \\
&\leq \mathbb{E}\left(\sum_{a \in \mathcal{A}}\left\{R_h(s,a) + \sum_{s' \in \mathcal{S}} P_h(s'|s,a)\bar{V}_{h+1}^*(s'|R)\right\}\right) \\
&= \sum_{a \in \mathcal{A}}\left(r_h(s,a) + \sum_{s' \in \mathcal{S}} P_h(s'|s,a)\mathbb{E}\left[\bar{V}_{h+1}^*(s'|R)\right]\right) \\
&\overset{(*)}{\leq} \sum_{a \in \mathcal{A}}\left(r_h(s,a) + A^{H-h}\sum_{s' \in \mathcal{S}} P_h(s'|s,a)V_{h+1}^{0,*}(s')\right) \\
&\leq A^{H-h}\sum_{a \in \mathcal{A}}\left(r_h(s,a) + \sum_{s' \in \mathcal{S}} P_h(s'|s,a)V_{h+1}^{0,*}(s')\right) \\
&\leq A^{H-h} \cdot A\max_{a \in \mathcal{A}}\left\{r_h(s,a) + \sum_{s' \in \mathcal{S}} P_h(s'|s,a)V_{h+1}^{0,*}(s')\right\} \\
&= A^{H+1-h}V_h^{0,*}(s),
\end{aligned}
$$

where throughout the derivation, we use the fact that all rewards (and thus the values) are non-negative and $(*)$ is due to the induction hypothesis. This concludes the proof of the lower bound in the statement, namely, that for all dynamics and rewards, it holds that $CR^H \geq \max\{\frac{1}{SAH}, \frac{1}{A^H}\}$.

**Worst-case environment – upper bound:** see Proposition 1, where we present a tree-like stationary environment for which the aforementioned bounds are near-tight. $\square$

# B    Proofs for Multi-Step Lookahead Agents

**Proposition 2.** *For any $L \in [H]$, let $t_L(h) = \max\{h - L + 1, 1\}$. Then, it holds that*

$$\sup_{\mathcal{R}^H \sim \mathcal{D}(r)} V^{L,*}(P, r) = \max_{\pi \in \Pi^{\mathcal{M}}} \sum_{(h,s,a) \in \mathcal{X}} r_h(s, a) \sum_{s' \in \mathcal{S}} d^{\pi}_{t_L(h)}(s') d^*_h(s | s_{t_L(h)} = s')$$

*Proof.* We start by lower-bounding the optimal value in the presence of long-shot rewards. Then, we prove a matching upper value for all rewards and $L$-step lookahead policies.

**Lower bound on the value of long-shots.** Let $\epsilon > 0$ and assume that $R \sim LS_\epsilon(r)$, namely, that for any $(h, s, a) \in \mathcal{X}$, a reward of $r_h(s, a)/\epsilon$ is generated with probability $\epsilon$; otherwise, the reward would be zero. Let $\pi \in \Pi^{\mathcal{M}}$ be any Markovian policy that does not observe future rewards and let $\tilde{\pi} \in \Pi^L$ be a policy that plays $\pi$ if all the $L$-step future rewards are zero and otherwise optimally navigates to one strictly positive reward (ties broken arbitrarily). In particular, if only one long-shot reward is realized at $(h, s, a)$, this policy would play $\pi$ until timestep $t_L(h) = \max\{h - L + 1, 1\}$ and then maximize the reaching probability from $s_{t_L(h)}$ to $s_h = s$. If the agent successfully reaches $s_h = s$, it will play $a_h = a$ and collect the reward.

The value of $\tilde{\pi}$ can be lower-bounded by the value that at most one long-shot is realized; Denoting

$$\mathbb{G}_{h,s,a} = \left\{ R_h(s, a = \frac{r_h(s, a)}{\epsilon}, \forall (h', s', a') \neq (h, s, a) : R_h(s, a) = 0 \right\},$$

the event that a reward was realized only in $(h, s, a) \in \mathcal{X}$, we bound

$$V^{L,*}(P, r) \geq V^{L,\tilde{\pi}}(P, r)$$

$$= \mathbb{E}\left[ \sum_{h'=1}^{H} R_{h'}(s_{h'}, a_{h'}) \Big| \tilde{\pi} \right]$$

$$\overset{(1)}{\geq} \sum_{(h,s,a) \in \mathcal{X}} \mathbb{E}\left[ \sum_{h'=1}^{H} R_{h'}(s_{h'}, a_{h'}) \Big| \tilde{\pi}, \mathbb{G}_{h,s,a} \right] \Pr\{\mathbb{G}_{h,s,a}\}$$

$$= \sum_{(h,s,a) \in \mathcal{X}} \mathbb{E}\left[ \frac{r_h(s, a)}{\epsilon} \mathbb{1}\{s_h = s, a_h = a\} \Big| \tilde{\pi}, \mathbb{G}_{h,s,a} \right] \Pr\{\mathbb{G}_{h,s,a}\}$$

$$\overset{(2)}{=} \sum_{(h,s,a) \in \mathcal{X}} \sum_{s' \in \mathcal{S}} \Pr\{s_{t_L(h)} = s' | \pi\} \max_{\pi' \in \Pi^{\mathcal{M}}} \Pr\{s_h = s, a_h = a | s_{t_L(h)} = s', \pi'\} \frac{r_h(s, a)}{\epsilon} \Pr\{\mathbb{G}_{h,s,a}\}$$

$$\overset{(3)}{=} \sum_{(h,s,a) \in \mathcal{X}} \sum_{s' \in \mathcal{S}} d^{\pi}_{t_L(h)}(s') d^*_h(s | s_{t_L(h)} = s') \frac{r_h(s, a)}{\epsilon} \Pr\{\mathbb{G}_{h,s,a}\}$$

$$\overset{(4)}{=} \sum_{(h,s,a) \in \mathcal{X}} \sum_{s' \in \mathcal{S}} d^{\pi}_{t_L(h)}(s') d^*_h(s | s_{t_L(h)} = s') \frac{r_h(s, a)}{\epsilon} \cdot \epsilon (1 - \epsilon)^{SAH-1}$$

$$\geq e^{-\epsilon SAH} \sum_{(h,s,a) \in \mathcal{X}} r_h(s, a) \sum_{s' \in \mathcal{S}} d^{\pi}_{t_L(h)}(s') d^*_h(s | s_{t_L(h)} = s').$$

In (1), we use the facts that the events $\mathbb{G}_{h,s,a}$ are disjoint and the rewards are non-negative. Next, in (2), we decompose to steps until $t_L(h)$, where we play $\pi$, and steps from $t_L(h)$ to $h$, where we try to maximize reaching probability to $(s, a)$ at timestep $h$. Notice that the reward is independent of the transition, so the optimal reaching policy is Markovian. Relation (3) replaces the probability notation to conditional occupancy measure and (4) substitutes the probability of the events. Maximizing over $\pi$ and taking the limit of small $\epsilon$, we get a lower bound of

$$\sup_{\mathcal{R}^H \sim \mathcal{D}(r)} V^{L,*}(P, r) \geq \sup_{\epsilon > 0} \max_{\pi \in \Pi^{\mathcal{M}}} e^{-\epsilon SAH} \sum_{(h,s,a) \in \mathcal{X}} r_h(s, a) \sum_{s' \in \mathcal{S}} d^{\pi}_{t_L(h)}(s') d^*_h(s | s_{t_L(h)} = s')$$

$$= \max_{\pi \in \Pi^{\mathcal{M}}} \sum_{(h,s,a) \in \mathcal{X}} r_h(s, a) \sum_{s' \in \mathcal{S}} d^{\pi}_{t_L(h)}(s') d^*_h(s | s_{t_L(h)} = s').$$

**Upper bound on the value of all reward distributions.** For any fixed lookahead policy $\pi \in \Pi^L$ and any reward distribution, we bound

$$
V^{L,\pi}(P,r) = \mathbb{E}\left[\sum_{h=1}^{H} R_h(s_h, a_h)|\pi\right]
$$

$$
= \sum_{(h,s,a)\in\mathcal{X}} \mathbb{E}[R_h(s,a)\mathbb{1}\{s_h = s, a_h = a\}|\pi]
$$

$$
= \sum_{(h,s,a)\in\mathcal{X}} \sum_{s'\in\mathcal{S}} \Pr\{s_{t_L(h)} = s'|\pi\}\mathbb{E}\big[R_h(s,a)\mathbb{1}\{s_h = s, a_h = a\}|\pi, s_{t_L(h)} = s'\big]
$$

$$
= \sum_{(h,s,a)\in\mathcal{X}} \sum_{s'\in\mathcal{S}} d^{\pi}_{t_L(h)}(s')\mathbb{E}\big[R_h(s,a)\Pr\{s_h = s, a_h = a|\pi, s_{t_L(h)} = s', R_h(s,a)\}|\pi, s_{t_L(h)} = s'\big]
$$

$$
\leq \sum_{(h,s,a)\in\mathcal{X}} \sum_{s'\in\mathcal{S}} d^{\pi}_{t_L(h)}(s')\mathbb{E}\left[R_h(s,a)\max_{\pi^*\in\Pi^L}\Pr\{s_h = s, a_h = a|\pi^*, s_{t_L(h)} = s', R_h(s,a)\}|\pi, s_{t_L(h)} = s'\right]
$$

$$
\overset{(1)}{=} \sum_{(h,s,a)\in\mathcal{X}} \sum_{s'\in\mathcal{S}} d^{\pi}_{t_L(h)}(s')d^{*}_h(s|s_{t_L(h)} = s')\mathbb{E}\big[R_h(s,a)|\pi, s_{t_L(h)} = s'\big]
$$

$$
\overset{(2)}{=} \sum_{(h,s,a)\in\mathcal{X}} \sum_{s'\in\mathcal{S}} d^{\pi}_{t_L(h)}(s')d^{*}_h(s|s_{t_L(h)} = s')r_h(s,a)
$$

$$
\leq \max_{\pi^*\in\Pi^L} \sum_{(h,s,a)\in\mathcal{X}} \sum_{s'\in\mathcal{S}} d^{\pi^*}_{t_L(h)}(s')d^{*}_h(s|s_{t_L(h)} = s')r_h(s,a)
$$

$$
\overset{(3)}{=} \max_{\pi^*\in\Pi^\mathcal{M}} \sum_{(h,s,a)\in\mathcal{X}} \sum_{s'\in\mathcal{S}} d^{\pi^*}_{t_L(h)}(s')d^{*}_h(s|s_{t_L(h)} = s')r_h(s,a)
$$

Relation (1) holds since the state dynamics are independent of the rewards realization and the maximal reaching probability is $d^{*}_h(s|s_{t_L(h)} = s')$. Relation (2) holds because we reach the state at timestep $t_L(h)$ *just before* seeing $R_h(s,a)$; therefore, the two variables are independent. Finally, relation (3) holds since we can rewrite the value as

$$
\max_{\pi^*\in\Pi^L} \sum_{i=1}^{H}\sum_{s'\in\mathcal{S}} d^{\pi^*}_i(s') \sum_{(h,s,a)\in\mathcal{X}} \mathbb{1}\{t_L(h) = i\}d^{*}_h(s|s_{t_L(h)} = s')r_h(s,a).
$$

This expression is equivalent to the optimal value of a no-lookahead agent whose expected reward at any $(i, s', a') \in \mathcal{X}$ is $\sum_{(h,s,a)\in\mathcal{X}} \mathbb{1}\{t_L(h) = i\}d^{*}_h(s|s_{t_L(h)} = s')r_h(s,a)$, so there exists a Markovian policy that maximizes this value. $\qquad\square$

**Theorem 5.** *[CR versus Multi-Step Lookahead Agents] For any $L \in [H]$, let $t_L(h) = \max\{h - L + 1, 1\}$. Then, it holds that*

*Worst-case distributions:* $CR^L(P, r) = \frac{\max_{\pi \in \Pi^\mathcal{M}} \sum_{(h,s,a) \in \mathcal{X}} r_h(s,a) d_h^\pi(s,a)}{\max_{\pi \in \Pi^\mathcal{M}} \sum_{(h,s,a) \in \mathcal{X}} r_h(s,a) \sum_{s' \in \mathcal{S}} d_{t_L(h)}^\pi(s') d_h^*(s|s_{t_L(h)} = s')}.$

*Worst-case reward expectations:*

$$CR^L(P) = \min_{\pi^* \in \Pi^\mathcal{M}} \max_{\pi \in \Pi^\mathcal{M}} \min_{(h,s,a) \in \mathcal{X}} \frac{d_h^\pi(s,a)}{\sum_{s' \in \mathcal{S}} d_{t_L(h)}^{\pi^*}(s') d_h^*(s|s_{t_L(h)} = s')}.$$

*If the reward expectations are stationary ($r_h(s,a) = r(s,a)$), then*

$$CR^L(P) = \min_{\pi^* \in \Pi^\mathcal{M}} \max_{\pi \in \Pi^\mathcal{M}} \min_{(s,a) \in \mathcal{S} \times \mathcal{A}} \frac{\sum_{h=1}^{H} d_h^\pi(s,a)}{\sum_{h=1}^{H} \sum_{s' \in \mathcal{S}} d_{t_L(h)}^{\pi^*}(s') d_h^*(s|s_{t_L(h)} = s')}.$$

*Worst-case environments: For all environments, $CR^L \geq \max\left\{ \frac{1}{SAH}, \frac{1}{(H-L+1)A^L} \right\}$. Also, for any $\delta \in (0,1)$ there exist stationary environments with rewards over $[0,1]$ s.t. if $S = A^n + 1$ for $n \in \{0, \dots, L-1\}$, then $CR^L(P,r) \leq \frac{1+\delta}{(H - \log_A(S-1)) \cdot (A-1)(S-1)}$, and if $S \geq A^L + 1$, then $CR^L(P,r) \leq \frac{1+\delta}{(H-L+1)(A^L-1)}$.*

*Proof.* **Worst-case distribution:** This part of the theorem is a directly corollary of Proposition 2, applied with Remark 2 and Equation (2). We remark that we assume w.l.o.g. that $r_h(s,a) > 0$ for at least one reachable $(h, s, a) \in \mathcal{X}$ (i.e., $d_h^*(s,a) > 0$). Otherwise, both values in the numerator and denominator equal zero and the ratio is defined as $+\infty$.

**Worst-case reward expectations:** As in the proof of Theorem 4, we start by rewriting the maximization problems in the competitive ratio using, $D^\mathcal{M}(P)$ the set of occupancy measures induced by the transition kernel $P$ and all Markovian policies:

$$CR^L(P) = \inf_{r_h \in [0,1]^{SA}} CR^L(P, r)$$

$$= \inf_{r_h \in [0,1]^{SA}} \frac{\max_{\pi \in \Pi^\mathcal{M}} \sum_{(h,s,a) \in \mathcal{X}} r_h(s,a) d_h^\pi(s,a)}{\max_{\pi^* \in \Pi^\mathcal{M}} \sum_{(h,s,a) \in \mathcal{X}} r_h(s,a) \sum_{s' \in \mathcal{S}} d_{t_L(h)}^{\pi^*}(s') d_h^*(s|s_{t_L(h)} = s')}$$

$$= \inf_{r_h \in [0,1]^{SA}} \min_{\pi^* \in \Pi^\mathcal{M}} \max_{\pi \in \Pi^\mathcal{M}} \frac{\sum_{(h,s,a) \in \mathcal{X}} r_h(s,a) d_h^\pi(s,a)}{\sum_{(h,s,a) \in \mathcal{X}} r_h(s,a) \sum_{s' \in \mathcal{S}} d_{t_L(h)}^{\pi^*}(s') d_h^*(s|s_{t_L(h)} = s')}$$

$$= \inf_{r_h \in [0,1]^{SA}} \min_{d' \in D^\mathcal{M}(P)} \max_{d \in D^\mathcal{M}(P)} \frac{\sum_{(h,s,a) \in \mathcal{X}} r_h(s,a) d_h(s,a)}{\sum_{(h,s,a) \in \mathcal{X}} r_h(s,a) \sum_{s' \in \mathcal{S}} d'_{t_L(h)}(s') d_h^*(s|s_{t_L(h)} = s')}$$

$$= \inf_{d' \in D^\mathcal{M}(P)} \inf_{r_h \in [0,1]^{SA}} \max_{d \in D^\mathcal{M}(P)} \frac{\sum_{(h,s,a) \in \mathcal{X}} r_h(s,a) d_h(s,a)}{\sum_{(h,s,a) \in \mathcal{X}} r_h(s,a) \sum_{s' \in \mathcal{S}} d'_{t_L(h)}(s') d_h^*(s|s_{t_L(h)} = s')}.$$

$$(8)$$

Continuing following the proof of Theorem 4, we use the convexity and compactness of the set of occupancy measures to apply Lemma 1 on the two internal problems, this time with $\alpha_{h,s,a} = \sum_{s' \in \mathcal{S}} d'_{t_L(h)}(s') d_h^*(s|s_{t_L(h)} = s')$. Doing so results with

$$CR^L(P) = \inf_{d' \in D^\mathcal{M}(P)} \max_{d \in D^\mathcal{M}(P)} \min_{(h,s,a) \in \mathcal{X}} \frac{d_h(s,a)}{\sum_{s' \in \mathcal{S}} d'_{t_L(h)}(s') d_h^*(s|s_{t_L(h)} = s')}.$$

At this point, we deviate from the previous proof and analyze the external optimization problem. In particular, we want to show that the minimum is obtained in the set of Markovian policies. We prove it using Lemma 2. For its application, notice that $D^\mathcal{M}(P)$ is a convex and compact polytope, and therefore so does its linear transformation $\mathcal{P} = \left\{ \sum_{s' \in \mathcal{S}} d'_{t_L(h)}(s') d_h^*(s|s_{t_L(h)} = s') | d \in D^\mathcal{M}(P) \right\}$, so the conditions of the lemma hold: the infimum is obtained at a minimizer in the set. Substituting it

back into $CR^L(P)$ and using the equivalence between occupancy measures and policies leads to the desired result:

$$CR^L(P) = \min_{d' \in D^{\mathcal{M}}(P)} \max_{d \in D^{\mathcal{M}}(P)} \min_{(h,s,a) \in \mathcal{X}} \frac{d_h(s,a)}{\sum_{s' \in \mathcal{S}} d'_{t_L(h)}(s') d_h^*(s|s_{t_L(h)} = s')} \quad \text{(Lemma 2)}$$

$$= \min_{\pi^* \in \Pi^{\mathcal{M}}} \max_{\pi \in \Pi^{\mathcal{M}}} \min_{(h,s,a) \in \mathcal{X}} \frac{d_h^\pi(s,a)}{\sum_{s' \in \mathcal{S}} d_{t_L(h)}^{\pi^*}(s') d_h^*(s|s_{t_L(h)} = s')}.$$

For stationary rewards, where $r_h(s,a) = r(s,a)$, we rewrite Equation (8) as

$$CR^L(P) = \inf_{d' \in D^{\mathcal{M}}(P)} \inf_{r_h \in [0,1]^{SA}} \max_{d \in D^{\mathcal{M}}(P)} \frac{\sum_{(s,a) \in \mathcal{S} \times \mathcal{A}} \left( \sum_{h=1}^H d_h(s,a) \right) r(s,a)}{\sum_{(s,a) \in \mathcal{S} \times \mathcal{A}} \left( \sum_{h=1}^H \sum_{s' \in \mathcal{S}} d'_{t_L(h)}(s') d_h^*(s|s_{t_L(h)} = s') \right) r(s,a)}.$$

We can now reapply Lemma 1 with the appropriate $\alpha_{s,a} = \sum_{h=1}^H \sum_{s' \in \mathcal{S}} d'_{t_L(h)}(s') d_h^*(s|s_{t_L(h)} = s')$, followed by applying Lemma 2, to get

$$CR^L(P) = \inf_{d' \in D^{\mathcal{M}}(P)} \max_{d \in D^{\mathcal{M}}(P)} \min_{(s,a) \in \mathcal{S} \times \mathcal{A}} \frac{\sum_{h=1}^H d_h(s,a)}{\sum_{h=1}^H \sum_{s' \in \mathcal{S}} d'_{t_L(h)}(s') d_h^*(s|s_{t_L(h)} = s')} \quad \text{(Lemma 1)}$$

$$= \min_{\pi^* \in \Pi^{\mathcal{M}}} \max_{\pi \in \Pi^{\mathcal{M}}} \min_{(s,a) \in \mathcal{S} \times \mathcal{A}} \frac{\sum_{h=1}^H d_h^\pi(s,a)}{\sum_{h=1}^H \sum_{s' \in \mathcal{S}} d_{t_L(h)}^{\pi^*}(s') d_h^*(s|s_{t_L(h)} = s')}. \quad \text{(Lemma 2)}$$

**Worst-case environment – lower bound:** First notice that by definition, any $L$-step lookahead policy is also a full lookahead policy. In particular, for all environments, $V^{L,*}(P,r) \leq V^{H,*}(P,r)$, and the reverse relation would hold for the CR. Thus, from Theorem 1, we directly get the lower bound $CR^L \geq CR^H \geq \frac{1}{SAH}$. We further proof that $CR^L \geq \frac{1}{(H-L+1)A^L}$ using a reduction to the full lookahead case.

To this end, we start by decomposing the no-lookahead value of any $\pi \in \Pi^{\mathcal{M}}$ as follows

$$V^{0,\pi}(P,r) = \sum_{(h,s,a) \in \mathcal{X}} r_h(s,a) d_h^\pi(s,a)$$

$$= \sum_{(h,s,a) \in \mathcal{X}} r_h(s,a) \Pr\{s_h = s, a_h = a\}$$

$$= \sum_{(h,s,a) \in \mathcal{X}} r_h(s,a) \sum_{s' \in \mathcal{S}} \Pr\{s_{t_L(h)} = s'\} \Pr\{s_h = s, a_h = a | s_{t_L(h)} = s'\}$$

$$= \sum_{(h,s,a) \in \mathcal{X}} r_h(s,a) \sum_{s' \in \mathcal{S}} d_{t_L(h)}^\pi(s') d_h^\pi(s,a|s_{t_L(h)} = s')$$

$$= \sum_{s' \in \mathcal{S}} d_1^\pi(s') \sum_{h=1}^L \sum_{(s,a) \in \mathcal{S} \times \mathcal{A}} r_h(s,a) d_h^\pi(s,a|s_1 = s')$$

$$+ \sum_{i=2}^{H-L+1} \sum_{s' \in \mathcal{S}} d_i^\pi(s') \sum_{(s,a) \in \mathcal{S} \times \mathcal{A}} r_{i+L-1}(s,a) d_{i+L-1}^\pi(s,a|s_h = s').$$

In the last inequality, we decompose the summation into two terms depending on whether $t_L(h) \triangleq i = 1$ or $t_L(h) > 1$. For brevity, let $\{r^i\}_{i=1}^{H-L+1}$ be such that for all $(h,s,a) \in \mathcal{X}$,

$$r_h^1(s,a) = r_h(s,a) \mathbb{1}\{h \in [L]\},$$

$$r_h^i(s,a) = r_h(s,a) \mathbb{1}\{h = i+L-1\}, \quad \forall i \in \{2, \dots H-L+1\}.$$

Using this notation, one could rewrite the value as

$$V^{0,\pi}(P,r) = \sum_{i=1}^{H-L+1} \sum_{s' \in \mathcal{S}} d_i^\pi(s') \sum_{(s,a) \in \mathcal{S} \times \mathcal{A}} \sum_{h=i}^{i+L-1} r_h^i(s,a) d_h^\pi(s,a|s_i = s'). \quad (9)$$

Notice that $r_h^i$ are the expected rewards of timesteps observed by the lookahead agent at step $i$. We now define the following set of policies

- A Markovian policy that maximizes the $L$-lookahead value is denoted by

$$\pi^* \in \arg\max_{\pi \in \Pi^{\mathcal{M}}} \sum_{(h,s,a) \in \mathcal{X}} r_h(s,a) \sum_{s' \in \mathcal{S}} d_{t_L(h)}^\pi(s') d_h^*(s|s_{t_L(h)} = s').$$

- For any $i \in [H - L + 1]$, let $\pi_i$ be a Markovian policy that plays $\pi^*$ until reaching some state $s_i$ and then continues by a policy that maximizes the reward function $r^i$ for the following $L$ timesteps:

$$\pi_i \in \arg\max_{\pi \in \Pi^{\mathcal{M}}} \sum_{(s,a) \in \mathcal{S} \times \mathcal{A}} \sum_{h=i}^{i+L-1} r_h^i(s,a) d_h^\pi(s,a|s_i).$$

For $i = 1$, the state $s_1$ would be the initial state, generated from the initial state distribution. Notice that starting for the $i^{th}$ timestep, $\pi_i$ is an optimal policy given rewards $r^i$ in the standard MDP model, so there exists an optimal Markovian policy that maximizes its value simultaneously for all $s_i \in \mathcal{S}$. By ignoring all but the $i^{th}$ term in Equation (9), one could clearly see that

$$V^{0,\pi_i} \geq \sum_{s' \in \mathcal{S}} d_i^{\pi^*}(s') \sum_{(s,a) \in \mathcal{S} \times \mathcal{A}} \sum_{h=i}^{i+L-1} r_h^i(s,a) d_h^{\pi_i}(s,a|s_i = s')$$

$$= \sum_{s' \in \mathcal{S}} d_i^{\pi^*}(s') \max_{\pi \in \Pi^{\mathcal{M}}} \sum_{(s,a) \in \mathcal{S} \times \mathcal{A}} \sum_{h=i}^{i+L-1} r_h^i(s,a) d_h^\pi(s,a|s_i = s').$$

- All aforementioned policies are Markovian, so by the convexity of the occupancies induced by Markovian policies [Altman, 2021], there exists $\pi_u \in \Pi^{\mathcal{M}}$ such that for all $(h, s, a) \in \mathcal{X}$,

$$d_h^{\pi_u}(s,a) = \frac{1}{H - L + 1} \sum_{i=1}^{H-L+1} d_h^{\pi_i}(s,a).$$

Since values are linear in the occupancy measure, we can bound the optimal no-lookahead value by

$$V^{0,*}(P, r) \geq V^{0,\pi_u}(P, r)$$

$$= \frac{1}{H - L + 1} \sum_{i=1}^{H-L+1} V^{0,\pi_i}$$

$$\geq \frac{1}{H - L + 1} \sum_{i=1}^{H-L+1} \sum_{s' \in \mathcal{S}} d_i^{\pi^*}(s') \max_{\pi \in \Pi^{\mathcal{M}}} \sum_{(s,a) \in \mathcal{S} \times \mathcal{A}} \sum_{h=i}^{i+L-1} r_h^i(s,a) d_h^\pi(s,a|s_i = s'). \quad (10)$$

Moving forwards, we use a similar decomposition to the $L$-lookahead value using Proposition 2:

$$V^{L,*}(P, r) \leq \max_{\pi \in \Pi^{\mathcal{M}}} \sum_{(h,s,a) \in \mathcal{X}} r_h(s,a) \sum_{s' \in \mathcal{S}} d_{t_L(h)}^\pi(s') d_h^*(s|s_{t_L(h)} = s')$$

$$= \sum_{(h,s,a) \in \mathcal{X}} r_h(s,a) \sum_{s' \in \mathcal{S}} d_{t_L(h)}^{\pi^*}(s') d_h^*(s|s_{t_L(h)} = s')$$

$$= \sum_{s' \in \mathcal{S}} d_1(s') \sum_{h=1}^{L} \sum_{(s,a) \in \mathcal{S} \times \mathcal{A}} r_h(s,a) d_h^*(s|s_1 = s')$$

$$+ \sum_{i=2}^{H-L+1} \sum_{s' \in \mathcal{S}} d_i^{\pi^*}(s') \sum_{(s,a) \in \mathcal{S} \times \mathcal{A}} r_{i+L-1}(s,a) d_{i+L-1}^*(s|s_i = s')$$

$$= \sum_{i=1}^{H-L+1} \sum_{s' \in \mathcal{S}} d_i^{\pi^*}(s') \sum_{(s,a) \in \mathcal{S} \times \mathcal{A}} \sum_{h=i}^{i+L-1} r_h^i(s,a) d_h^*(s|s_i = s'). \quad (11)$$

To conclude the reduction, recall the inequality $\frac{\sum_i \alpha_i x_i}{\sum_i \alpha_i y_i} \geq \min_i \left\{ \frac{x_i}{y_i} \right\}$, which holds for all values of $x_i, y_i, \alpha_i \geq 0$ s.t. $\sum_i \alpha_i > 0$, due to the quasiconcavity of the ratio of linear functions (given the convention that $x/0 = +\infty$). Applying this on the CR with the coefficients $\alpha_{i,s'} = d_i^{\pi^*}(s')$ and using Equations (10) and (11), we get for all environments that

$$CR^L(P, r) = \frac{V^{0,*}(P, r)}{V^{L,*}(P, r)}$$

$$\geq \frac{1}{H - L + 1} \min_{\substack{i \in [H-L+1], \\ s' \in \mathcal{S}}} \left\{ \frac{\max_{\pi \in \Pi^{\mathcal{M}}} \sum_{(s,a) \in \mathcal{S} \times \mathcal{A}} \sum_{h=i}^{i+L-1} r_h^i(s, a) d_h^\pi(s, a | s_i = s')}{\sum_{(s,a) \in \mathcal{S} \times \mathcal{A}} \sum_{h=i}^{i+L-1} r_h^i(s, a) d_h^*(s | s_i = s')} \right\}$$

$$\overset{(*)}{\geq} \frac{1}{(H - L + 1)A^L}.$$

The last inequality is the reduction to the full lookahead: each of the terms is exactly the CR versus a full lookahead agent with horizon $L$ and reward expectations $r^i$ (see Theorem 1, part 1). Thus, each of the terms is lower-bounded by the bound for the worst-case environment given horizon $L$ (see Theorem 1, part 3) – by $\frac{1}{A^L}$.

**Worst-case environment – upper bound:** as in Theorem 3, this part of the proof is covered in Proposition 1, where we present a tree-like stationary environment with the stated behavior. $\square$

## C Analyzing the Competitive Ratio of Specific Environments

### C.1 Upper-Bounds for Reward Lookahead – Delayed Trees

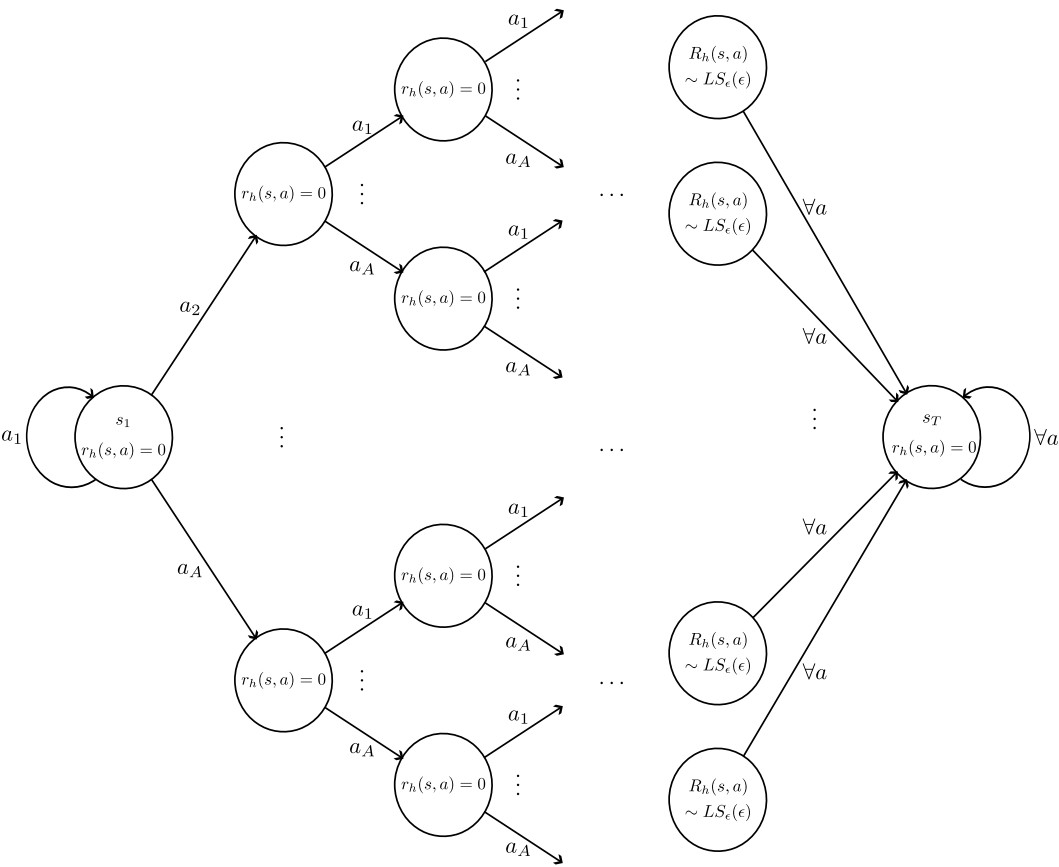

Figure 2: A near-worst-case environment: tree-like MDP. An agent can decide to stay at the root of the tree, but once it starts to traverse the tree, it must navigate to one of its leaves, from which it moves to a non-rewarding terminal state. All leaves have long-shot rewards, while all other nodes yield no reward.

**Proposition 1.** *For any $L \in [H]$ and any $\delta \in (0, 1)$, there exist stationary environments with rewards over $[0, 1]$ s.t. if $S = A^n + 1$ for $n \in \{0, \ldots, L - 1\}$, then $CR^L \leq \frac{1+\delta}{(H - \log_A(S-1)) \cdot (A-1)(S-1)}$, and if $S \geq A^L + 1$, then $CR^L \leq \frac{1+\delta}{(H-L+1)(A^L-1)}$. Moreover, if $L = H$ and $S \geq A^H - 1$, there exists an environment s.t. $CR^H \leq \frac{1+\delta}{A^H}$*

*Proof.* Assume that $S = A^n + 1$ for some $n \in \mathbb{N}$. We divide the proof into different cases, depending on the values of $n$ and $L$.

**Case 1:** $L \in [H]$ and $n \in [1, L - 1]$. To prove this bound, we design a tree MDP with an additional option to decide *when* to traverse it, as illustrated in Figure 2. In particular, assume that the fixed initial state $s_1$ is the root of a tree of depth $n + 1$ such that the root has $A - 1$ descendants and all other nodes have $A$ descendants. Thus, the number of nodes in this tree is

$$1 + (A - 1) \sum_{i=1}^{n} A^{i-1} = 1 + (A - 1) \frac{A^n - 1}{A - 1} = A^n,$$

and the number of leaves is $(A - 1)A^{n-1}$. Assuming that after traversing the tree, the environment moves to a terminal state $s_T$, this environment could indeed be represented using $S = A^n + 1$ states. We denote the dynamics of this tree by $P$.

For the dynamics, we allocate one action in the root of the tree that keeps the agent at the root, while the rest of the actions allow traversing through the tree. At the leaves, all actions transition to a terminal state $s_T$. We emphasize that once an agent has decided to start traversing the tree, it has to continue all the way until the leaves (and terminal state), so the decision when to traverse the tree is taken at its root. Finally, the reward of any action at any leaf is a long-shot $LS_\epsilon(\epsilon)$, namely Bernoulli-distributed with probability $\epsilon$. In particular, this distribution is bounded in $[0, 1]$.

In this example, any agent with no lookahead information will perform at most one action at a single leaf, independently of the reward realization, thus collecting in expectation no more than the expected reward of a single leaf $V^{0,*} \leq \mathbb{E}[LS_\epsilon(\epsilon)] = \epsilon$.

On the other hand, an $L$-lookahead agent could start traversing the tree only when a reward will be realized upon its arrival to the leaf. To reach a leaf at timestep $h$, the agent has to start traversing the tree at timestep $h - n$. Thus, this agent will wait at the root to see whether a reward is realized in any leaf at timesteps $\{n + 1, \ldots, H\}$, and only if so, will traverse it.

Since there are $(A - 1)A^{n-1}$ leaves with $A$ actions each, the probability that no reward is realized in any leaf at these timesteps is $(1 - \epsilon)^{(H-n)\cdot(A-1)A^{n-1}\cdot A}$, and the optimal lookahead agent would collect an expected reward of at least

$$V^{L,*} \geq 1 - (1 - \epsilon)^{(H-n)\cdot(A-1)A^n}$$
$$\geq (H - n) \cdot (A - 1)A^n\epsilon - ((H - n) \cdot (A - 1)A^n)^2\epsilon^2,$$

where the last inequality is since $(1 - x)^n \leq 1 - nx + n^2x^2$.

Combining both inequalities, for this environment we have that

$$CR^L(P, r) = \frac{V^{0,*}}{V^{L,*}}$$
$$\leq \frac{\epsilon}{(H - n) \cdot (A - 1)A^n\epsilon - ((H - n) \cdot (A - 1)A^n)^2\epsilon^2}$$
$$= \frac{1}{(H - n) \cdot (A - 1)A^n - ((H - n) \cdot (A - 1)A^n)^2\epsilon}.$$

In particular, for any $\delta > 0$, we could fix $\epsilon$ small enough such that

$$CR^L(P, r) \leq \frac{1 + \delta}{(H - \log_A(S - 1)) \cdot (A - 1)(S - 1)},$$

where we used the relation $S = A^n + 1$.

**Case 2:** $L \in [H]$ and $n = 0$. This is the case of $S = 2$. We separate it for the clarity of presentation, but the example remains the same: the first state $s_1$ is the initial state and the second $s_2 = s_T$ is a terminal non-rewarding state. When in $s_1$, a single action does not change the state but yields no reward, while all other $A - 1$ actions transition the environment to state $s_2$, giving a long-shot reward $LS_\epsilon(\epsilon)$. As in the first case, without any lookahead information, the agent could collect a reward at most once and obtain in expectation at most $V^{0,*} \leq \mathbb{E}[LS_\epsilon(\epsilon)] = \epsilon$.

On the other hand, any lookahead agent would move from $s_1$ to $s_2$ only when a reward is realized. Since there are $A - 1$ rewarding actions and $H$ opportunities to collect rewards, a lookahead agent could collect at least

$$V^{L,*} \geq 1 - (1 - \epsilon)^{H(A-1)} \geq H(A - 1)\epsilon - H^2(A - 1)^2\epsilon^2,$$

where the last inequality is again due to the inequality $(1 - x)^n \leq 1 - nx + n^2x^2$.

Combining both bounds, we now get

$$CR^L(P, r) = \frac{V^{0,*}}{V^{L,*}} \leq \frac{\epsilon}{H(A - 1)\epsilon - H^2(A - 1)^2\epsilon^2} = \frac{1}{H(A - 1) - H^2(A - 1)^2\epsilon}.$$

Thus, for any $\delta > 0$, there exist small enough $\epsilon$ such that

$$CR^L(P, r) \leq \frac{1 + \delta}{H(A - 1)}.$$

**Case 3:** $L \in [H-1]$ and $S \geq A^{L-1}+1$. We use the same example as in the first case and $n = L-1$, ignoring all extra states. Direct substitution to that bound results with

$$CR^L(P,r) \leq \frac{1+\delta}{(H-L+1)\cdot(A-1)(S-1)},$$

**Case 4:** Finally, if $L = H$ and $S \geq A^H - 1$, we discard the loop at the root and just build a full tree of depth $H$, leading to $A^{H-1}$ leaves (with $A$ actions each). From the root, the full lookahead agent can reach any leaf with a realized reward, which exists with probability $1 - (1-\epsilon)^{A^H}$. Following the exact same analysis would now yield $CR^H(P,r) \leq \frac{1+\delta}{A^H}$ for any $\delta > 0$, concluding the proof.

**Modification when $S \in [A^n + 2, A^{n+1}]$:** In this case, we cannot build a complete tree of depth $\log_A(S-1) + 1$. Instead, we start from the complete tree of depth $\lfloor \log_A(S-1) \rfloor + 1$ and use any extra states to create additional leaves of depth $\lceil \log_A(S-1) \rceil + 1$. The number of leaves for $S = A^n + 1$ was $N_0 = (A-1)A^{n-1}$. Therefore, $\lfloor \frac{S-N_0}{A} \rfloor$ of these leaves will have $A$ descendants in the new tree, increasing the number of leaves by $A-1$ each, while one additional 'old' leaf will take the rest of the states. For this reason, the total number of leaves $N$ would be

$$N \geq N_0 + (A-1)\left\lfloor \frac{S-N_0}{A} \right\rfloor + \left(S - N_0 - A\left\lfloor \frac{S-N_0}{A} \right\rfloor - 1\right)$$

$$= S - \left\lfloor \frac{S-N_0}{A} \right\rfloor - 1$$

$$\geq S\left(1 - \frac{1}{A}\right) - 2.$$

Recalling that in each leaf, we have $A$ possible actions, so rewards could be realized in $NA$ locations, and increasing the depth by 1 (so that the lookahead agent has one less attempt), we can follow the exact same analysis and get a more general bound of

$$CR^L(P,r) \leq \frac{1+\delta}{(H - \lceil \log_A(S-1) \rceil)\cdot(S(A-1)-2A)} = \Theta\left(\frac{1+\delta}{(H - \log_A(S-1))\cdot AS}\right)$$

for any $A + 2 \leq S \leq A^L + 1$. $\qquad\qquad\qquad\qquad\qquad\qquad\qquad\qquad\qquad\qquad\qquad\qquad\quad\square$

### C.2 Analysis of Grid MDPs

In the grid MDP, an agent starts at the bottom-left corner of an $n \times n$ grid and can either move up or right until getting to the top-right corner ('Manhattan navigation', see Figure 1(b)). After taking one last action, the interaction ends. We denote the states on the $i^{th}$ column (starting from the left) by $s_{i,1}, \ldots, s_{i,n}$ (with $s_{i,1}$ as the bottom state) and the states on the $j^{th}$ row (starting from the bottom) by $s_{1,j}, \ldots, s_{n,j}$ (with $s_{1,j}$ as the leftmost state). At the top edge of the grid, the agent must move right, and at the right edge, it must move up. The size of the state space is $S = n^2$, the action space is of size (at most) $A = 2$ and the horizon is $H = 2n - 1$.

This MDP generalizes the chain MDP with $A = 2$, analyzed in Section 5; indeed, by setting the reward to be non-zero only when going up from the bottom row ($s_{1,j}$), we effectively get a chain of length $n$ and a corresponding CR of $CR^H = CR^1 = \frac{1}{n} = \frac{2}{H+1}$. In particular, the reduction immediately leads to an upper bound of $\mathcal{O}(\frac{1}{H})$ for $CR^1$ (and $CR^H$), where the bound for one-step is almost worst-case, since $CR^1 \geq \frac{1}{AH} = \frac{1}{2H}$. Interestingly, this is a near-worst-case reward placement also versus full lookahead for the grid-MDP, as we now prove.

One way to prove this is to analyze a *flow* on the grid graph, which is equivalent to occupancy in deterministic MDPs. The value of the full lookahead agent corresponds with the maximal possible flow through any edge in the graph, which is the unit flow ($d^*_{i+j-1}(s_{i,j}) = 1$). Hence, the goal of the no-lookahead agent is to make sure that there is a minimal flow in all the edges of the graph, and this minimum would be the CR. This could be achieved by distributing a flow on the bottom and leftmost states and sending it in straight lines to the other side of the grid, as explained in Figure 3. The resulting flow ensures a minimal flow of $\frac{1}{2(n-1)}$ through all the edges. Even more, looking at the flow description, we could explicitly write the stochastic policy that achieves this flow by looking at

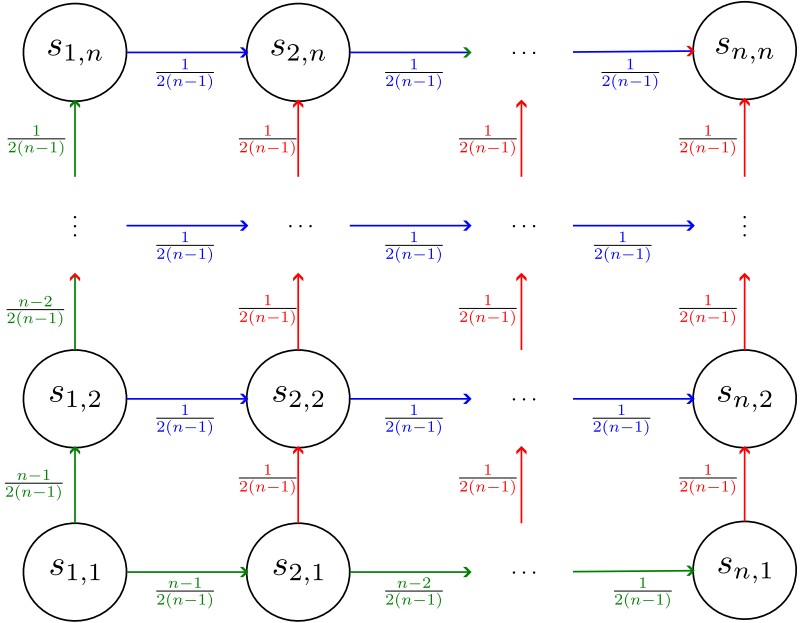

Figure 3: Illustration of a possible flow on a grid graph, starting from the bottom-left corner and ending at the top-right corner. The first step is to distribute the flow on the bottom and leftmost states, such that there is excess flow of $\frac{1}{2(n-1)}$ flow in each of these states (green). At the leftmost state, this excess flow is sent at a direct line towards the right (blue), while in the bottom row, this flow is sent up (red). Such flow ensures that all edges have a minimal flow of $\frac{1}{2(n-1)}$.

the ratio of the flow in each direction:

$$\pi(\text{Move-Right}|s_{i,j}) = \begin{cases} \frac{1}{2} & i = j = 1 & \text{(start)} \\ \frac{n-i}{n-i+1} & i = 1, j \in \{2, \ldots, n-1\} & \text{(bottom)} \\ \frac{1}{n-j+1} & j = 1, i \in \{2, \ldots, n-1\} & \text{(leftmost)} \\ \frac{1}{2} & i, j \in \{2, \ldots, n-1\} & \text{(middle)} \\ 1 & i = n, j \in \{2, \ldots, n-1\} & \text{(top)} \\ 0 & j = n, i \in \{2, \ldots, n-1\} & \text{(rightmost)} \\ \frac{1}{2} & i = j = n & \text{(end)} \end{cases}$$

For this policy, it is easy to prove that the minimal occupancy $d_{i+j-1}^{\pi}(s_{i,j}, a)$ is lower-bounded by $\frac{1}{2(n-1)}$ by directly verifying on the edges of the grid (starting from the bottom and left edges and then continuing to the top and right ones), and then proving with a simple induction that strictly inside the grid, $d_{i+j-1}^{\pi}(s_{i,j}) = \frac{1}{n-1}$. This implies that

$$CR^H(P) = \max_{\pi \in \Pi^{\mathcal{M}}} \min_{(h,s,a) \in \mathcal{X}} \frac{d_h^{\pi}(s,a)}{d_h^*(s)} = \max_{\pi \in \Pi^{\mathcal{M}}} \min_{(i.j) \in [n]^2, a \in \mathcal{A}} d_{i+j-1}^{\pi}(s_{i,j}, a) \geq \frac{1}{2(n-1)} = \frac{1}{H-1}.$$

In particular, for the grid MDP, the worst-case CR for full lookahead is at most worse by a factor of 2 compared to the CR versus one-step lookahead, similar to the chain MDP. However, in contrast to chains, where the prophet inequality ensures a constant ratio between one-step and full lookahead, in grids, this ratio could depend on $H$. For example, assume long-shot rewards $R \sim LS_\epsilon(1)$ for arbitrarily small $\epsilon$. As we already calculated the value for long-shot rewards, we know that one-step lookahead agents effectively collect all expected rewards along their trajectory (Equation (4)) – at most $2H$ rewards – while the full lookahead agents collect all reachable rewards (Equation (6)) – a total of $\Omega(H^2)$ rewards. At first glance, it might be seen as a contradiction, following a logic that

$$\text{„}\frac{\text{no-lookahead}}{\text{full lookahead}} = \frac{\text{no-lookahead}}{\text{one-step lookahead}} \cdot \frac{\text{one-step lookahead}}{\text{full lookahead}}\text{„},$$

but the careful reader would notice that the CRs are derived for very different reward expectations; one CR is calculated for sparse chain-like rewards while the other is calculated for dense rewards where all expectations are equal.

**Dense rewards.** We end this example by analyzing the CR when rewards are dense – all rewards are of unit expectation. Since all reward expectations are equal to 1, regardless of the policy, the value of all no-lookahead agents would trivially be $H$. For the value of $L$-lookahead agents we use Proposition 2 and rewrite the value by decomposing to different values of $t_L(h)$ as follows:

$$
\sup_{\mathcal{R}^H \sim \mathcal{D}(1)} V^{L,*}(P,1) = \max_{\pi \in \Pi^{\mathcal{M}}} \sum_{(h,s,a)\in\mathcal{X}} 1 \cdot \sum_{s'\in\mathcal{S}} d_{t_L(h)}^{\pi}(s') d_h^*(s|s_{t_L(h)} = s')
$$

$$
= \sum_{(s,a)\in\mathcal{S}\times\mathcal{A}} \sum_{h=1}^{L} d_h^*(s) + \max_{\pi\in\Pi^{\mathcal{M}}}\left\{ \sum_{t=2}^{H-L+1} \sum_{s'\in\mathcal{S}} d_t^{\pi}(s') \sum_{(s,a)\in\mathcal{S}\times\mathcal{A}} d_{t+L-1}^*(s|s_t = s') \right\}
$$

Since the environment is deterministic, all occupancies $d^*$ are binary: one if a state is reachable and zero otherwise. From the initial state, there are $L^2$ reachable states so the first term is equal to $L^2$. For the second term, we bound the number of reachable states after exactly $L$ steps by $L+1$ (all the possible number of 'up' moves between 0 and $+L$). This yields the bound

$$
\sup_{\mathcal{R}^H \sim \mathcal{D}(1)} V^{L,*}(P,1) \le L^2 + \max_{\pi\in\Pi^{\mathcal{M}}}\left\{ \sum_{t=2}^{H-L+1} \underbrace{\sum_{s'\in\mathcal{S}} d_t^{\pi}(s')}_{=1}(L+1) \right\}
$$

$$
= L^2 + (H-L)(L+1)
$$

$$
\le H(L+1)
$$

and result with a CR of $CR^L(P,1) \ge \frac{1}{L+1}$.

This bound is near-tight, again using Proposition 2. For the proof, we focus on a policy $\pi$ that iterates between moving up and right. As previously explained, the number of reachable states when looking $L$ steps forward is $L+1$ if we could perform all combinations of moving up and right. In particular, this is the case as long as we are not too close to the top-right border of the grid. By iterating the movements upwards and rightwards, for any $h \le H - 2L$, we arrive to a state $s_{i,j}$ such that $\max\{i,j\} \le \lceil \frac{h}{2} \rceil \le \lceil \frac{2n-1-2L}{2} \rceil \le n - L$, which ensures we are a distance of at least $L$ from the border. Therefore, we can bound

$$
\sup_{\mathcal{R}^H \sim \mathcal{D}(1)} V^{L,*}(P,1) \ge L^2 + \sum_{t=2}^{H-2L} \underbrace{\sum_{s'\in\mathcal{S}} d_t^{\pi}(s')}_{=1}(L+1)
$$

$$
= L^2 + \max\{(H-2L-1)(L+1), 0\}
$$

$$
= \Omega(HL),
$$

where the last relation is immediately obtained by looking at either $L < H/4$ or $L \ge H/4$. Thus, we also have that $CR^L(P,1) = \mathcal{O}(1/L)$, and we can conclude that $CR^L(P,1) = \Theta(1/L)$.

### C.3   Upper-Bound for Transition Lookahead

In this appendix, we analyze the competitive ratio versus one-step transition lookahead agents. Formally, at each timestep $h$ and state $s_h$, such agents observe what the next state $s_{h+1}$ would be upon playing any of the actions $a \in \mathcal{A}$. We assume that this is the only information available to the agent (namely, the agent has no reward lookahead). We also assume that transitions are generated independently at different timesteps and are independent of the rewards. Notably, even with one-step information, the CR is exponentially small, as stated in the following proposition:

**Proposition 2.** *For any $A \ge 2$, $H \ge 5$ and $S \ge A^{\left(1-\frac{1}{e}\right)H}$, there exists an environment such that the CR versus one-step transition lookahead agents is $CR \le \frac{2}{(A-1)^{\left(1-\frac{1}{e}\right)H-3}}$.*

*Proof.* The environment we build is a complete tree of depth $d$ (to be determined), where each node has $A - 1$ descendants. The agent always starts at the root of the tree. At each node, the agent can play $A = 1$ to stay at the same node, while the rest of the $A - 1$ actions move the agent to one of the descendants of the node uniformly at random. Only one leaf has a deterministic unit reward of $R = 1$ for all actions, while all other leaves yield no reward. After traversing the tree, the agent moves to a terminal non-rewarding state $s_T$. The total number of states required to create this environment is

$$1 + \sum_{i=0}^{d-1}(A-1)^i \le 2 + (A-1)\sum_{i=0}^{d-2} A^i = A^{d-1} + 1 \le A^d,$$

and the number of leaves in the tree is $N = (A-1)^{d-1}$. A no-lookahead agent could not do better than randomly traversing the tree and would obtain an expected reward of at most $V^{0,*} \le 1/N$.

On the other hand, one-step transition lookahead agents could choose the following policy: if an action leads in the direction of the rewarding leaf, take it; otherwise, wait in the current node. To obtain the reward, there have to be at least $d - 1$ timesteps where an action leads in the right direction, over the span of $H - 1$ attempts (one additional round is required to collect the reward. Letting $p_s = 1 - \left(1 - \frac{1}{A-1}\right)^{A-1} \ge 1 - \frac{1}{e}$ be the probability that such an action exists at a certain node ('success'), the value of the one-step lookahead agent would be at least the probability that a binomial distribution $\text{Bin}(n = H - 1, p = p_s)$ has at least $d - 1$ successes. Setting $d = \left\lfloor \left(1 - \frac{1}{e}\right)H \right\rfloor - 1$, so that $d - 1 \le \left(1 - \frac{1}{e}\right)(H - 1) - 1$, we use Hoeffding's inequality to get

$$V^{1,*} \ge \Pr\left(\text{Bin}\left(H - 1, 1 - \frac{1}{e}\right) \ge d - 1\right)$$

$$\ge 1 - \exp\left(-2\left(\left(1 - \frac{1}{e}\right)(H - 1) - (d - 1)\right)^2\right)$$

$$\ge 1 - \frac{1}{e^2}.$$

Therefore, the competitive ratio is upper-bounded for

$$CR \le \frac{1/(A-1)^{d-1}}{1 - \frac{1}{e^2}} \le \frac{2}{(A-1)^{\left(1 - \frac{1}{e}\right)H - 3}}.$$

We remark that the constraint $A \ge 2$ allows building such a tree, while $H \ge 5$ ensures a depth of at least $d \ge 2$. $\qquad\square$

## D Auxiliary Lemmas

**Lemma 1.** *Let $d \in \mathbb{N}$ and $\alpha \in \mathbb{R}_+^d$. Also, let $D \subset \mathbb{R}_+^d$ be a convex compact nonempty set. Then,*

$$\inf_{x \in [0,1]^d} \max_{y \in D} \frac{y^T x}{\alpha^T x} = \max_{y \in D} \min_{i \in [d]} \frac{y_i}{\alpha_i},$$

*where we define all ratios to be $+\infty$ if the denominator equals zero.*

*Proof.* We first remark that if $\alpha_i = 0$ for all $i \in [d]$, then by the definition of the division by zero, both sides are trivially equal to $+\infty$, and the result holds. Thus, from this point onwards, we assume w.l.o.g. that for some $i_0 \in [d]$, it holds that $\alpha_{i_0} > 0$.

**Step I:** We start from analyzing the l.h.s. problem and showing that

$$\inf_{x \in [0,1]^d} \max_{y \in D} \frac{y^T x}{\alpha^T x} = \inf_{\substack{z \in \mathbb{R}_+^d \\ \alpha^T z = 1}} \max_{y \in D} y^T z.$$

Notice that choosing $x_i = \mathbb{1}\{i = i_0\}$ leads to a bounded value of $\frac{\max_{y \in D} y_{i_0}}{\alpha_{i_0}} < \infty$, so the value is finite – there cannot be a solution such that $\alpha^T x = 0$ (and the value is $+\infty$), and we can w.l.o.g add the constraint $\alpha^T x > 0$. We further remark that both the numerator and denominator are always non-negative, so the infimum is bounded from below by $0$. Given that, the internal problem is always well-defined, and the maximizer is given by $y_x \in \arg\max_{y \in D} y^T x$.

We next show that the constraints $x \in [0,1]^d, \alpha^T x > 0$ can be replaced by the constraints $x \in \mathbb{R}_+^d, \alpha^T x = 1$. First, for any $x \in [0,1]^d$ s.t. $\alpha^T x = 1$, define $z_x = \frac{x}{\alpha^T x} \in \mathbb{R}_+^d$, for which $\alpha^T z = 1$ and

$$\max_{y \in D} \frac{y^T x}{\alpha^T x} = \max_{y \in D} y^T \frac{x}{\alpha^T x} = \max_{y \in D} y^T z_x \geq \inf_{\substack{z \in \mathbb{R}_+^d \\ \alpha^T z = 1}} \max_{y \in D} y^T z.$$

Thus, we have the inequality

$$\inf_{x \in [0,1]^d} \max_{y \in D} \frac{y^T x}{\alpha^T x} = \inf_{\substack{x \in [0,1]^d \\ \alpha^T x > 0}} \max_{y \in D} \frac{y^T x}{\alpha^T x} \geq \inf_{\substack{z \in \mathbb{R}_+^d \\ \alpha^T z = 1}} \max_{y \in D} y^T z.$$

On the other hand, for any $z \in \mathbb{R}_+^d$ s.t. $\alpha^T z = 1$, define $x_z = \frac{z}{\max_i z_i}$ (which is well defined due to the constraints). For this choice, we get that $x_z \in [0,1]^d$ and $\alpha^T x_z = \frac{\alpha^T z}{\max_i z_i} = \frac{1}{\max_i z_i} > 0$. In particular, one can write $z = \frac{x_z}{\alpha^T x_z}$, which implies that

$$\max_{y \in D} y^T z = \max_{y \in D} y^T \frac{x_z}{\alpha^T x_z} = \max_{y \in D} \frac{y^T x_z}{\alpha^T x_z} \geq \inf_{x \in [0,1]^d} \max_{y \in D} \frac{y^T x}{\alpha^T x}.$$

Therefore, we also have the other inequality

$$\inf_{\substack{z \in \mathbb{R}_+^d \\ \alpha^T z = 1}} \max_{y \in D} y^T z \geq \inf_{x \in [0,1]^d} \max_{y \in D} \frac{y^T x}{\alpha^T x},$$

which implies equality

$$\inf_{x \in [0,1]^d} \max_{y \in D} \frac{y^T x}{\alpha^T x} = \inf_{\substack{z \in \mathbb{R}_+^d \\ \alpha^T z = 1}} \max_{y \in D} y^T z.$$

**Step II:** Applying the minimax theorem.

The objective is linear in $z, y$ (and thus convex and concave in the variables, respectively), and the set $D$ is convex and compact. The constraint on $z$ is also convex, though not compact, but this is easily fixable; notice that for all $i$ such that $\alpha_i = 0$, $z_i$ does not affect the constraint. On the other

hand, setting $z_i > 0$ can only increase the objective since $y_i, z_i \geq 0$. Indeed, for any $z \in \mathbb{R}_+^d$ s.t. $\alpha^T z = 1$, letting $\tilde{z}_i = z_i \mathbb{1}\{\alpha_i > 0\}$, we have $\alpha^T \tilde{z} = 1$ and $y^T z \leq y^T \tilde{z}$. Hence, w.l.o.g., we can always add the constraint that $z_i = 0$ for all $i \in [d]$ with $\alpha_i = 0$. With this additional constraint, the set $\mathcal{Z} = \{z \in \mathbb{R}_+^d | \alpha^T z = 1, \forall i \text{ s.t. } \alpha_i = 0 : z_i = 0\}$ is convex and compact, so the infimum is actually a minimum and we can apply the minimax theorem to obtain

$$\inf_{\substack{z \in \mathbb{R}_+^d \\ \alpha^T z = 1}} \max_{y \in D} y^T z = \max_{y \in D} \min_{z \in \mathcal{Z}} y^T z$$

**Step III:** Solving the internal problem for fixed values of $y$.

At this point, we note that components where $\alpha_i = 0$ do not affect either the value or the solution. Therefore, from this point onwards, we assume w.l.o.g that $\alpha_i > 0$ for all $i$; we will then apply our results only on the subset of components with $\alpha_i > 0$. Given that, we also assume w.l.o.g. that $y_i > 0$ for all $i$ – otherwise, the constraint could be met by letting $z_i > 0$ for components with $y_i = 0$, which would lead to the optimal value of $0$ (we verify this case at the end of the proof).

Thus, we focus on solving the following problem: for any fixed $y \in \mathbb{R}^d$ s.t. $y_i > 0$ for all $i$, solve

$$\begin{aligned} \min_z \quad & y^T z \\ s.t. \quad & z_i \geq 0, \ \forall i \in [d], \\ & \alpha^T z = 1. \end{aligned}$$

Due to the linearity of both the objective and constraints (in $z$), KKT conditions are both necessary and sufficient for the solution of this problem. Letting $\mu$ and $\lambda$ be the dual variables for the constraints $z \in \mathbb{R}_+^d$ and $\alpha^T z = 1$, respectively, the KKT requires that for all $i \in [d]$,

$$\begin{aligned} y_i - \mu_i - \lambda \alpha_i &= 0 && \text{(stationarity)} \\ \mu_i z_i &= 0 && \text{(complementary slackness)} \\ \mu_i \geq 0, \ z_i &\geq 0 && \text{(feasibility 1)} \\ \alpha^T z &= 1. && \text{(feasibility 2)} \end{aligned}$$

For the stationarity to hold with the non-negativity of $\mu_i$, we must have that $\lambda \leq \min_{i \in [d]} \frac{y_i}{\alpha_i}$. Moreover, if this is a strict inequality, all $\mu_i$ are strictly positive, which leads to the infeasible zero-reward vector (due to the complementary slackness). Therefore, we can conclude that $\lambda = \min_{i \in [d]} \frac{y_i}{\alpha_i}$, and so $\mu_i = 0$ only in coordinates where this minimal ratio in achieved. By complementary slackness, $z_i = 0$ for the rest of the coordinates.

Substituting in the equality constraint, we get

$$1 \overset{(1)}{=} \sum_{i=1}^d \alpha_i z_i \overset{(2)}{=} \sum_{i : \frac{y_i}{\alpha_i} = \lambda} \alpha_i z_i = \sum_{i : \frac{y_i}{\alpha_i} = \lambda} \frac{y_i}{\lambda} z_i \overset{(2)}{=} \frac{1}{\lambda} \sum_{i=1}^d y_i z_i.$$

Explicitly, (1) is by the constraint and (2) is since $z_i = 0$ when $\frac{y_i}{\alpha_i} > \lambda$. Reorganizing, we get that the value of the internal problem is

$$\sum_{i=1}^d y_i z_i = \lambda = \min_{i \in [d]} \frac{y_i}{\alpha_i}.$$

We end by remarking that when $y_i = 0$ for some $i \in [d]$, the value becomes $0$ so that the result also holds in this case.

**Summary:** Combining all parts of the proof, we got

$$\inf_{x \in [0,1]^d} \max_{y \in D} \frac{y^T x}{\alpha^T x} = \max_{y \in D} \min_{i : \alpha_i > 0} \frac{y_i}{\alpha_i}$$

If we define the internal value to be $+\infty$ when $\alpha_i = 0$, we can further write

$$\inf_{x \in [0,1]^d} \max_{y \in D} \frac{y^T x}{\alpha^T x} = \max_{y \in D} \min_{i \in [d]} \frac{y_i}{\alpha_i},$$

which concludes the proof.

**Remark 3.** *Following almost identical proof, we could similarly prove that*

$$\inf_{x \in \mathbb{R}_+^d} \max_{y \in D} \frac{y^T x}{\alpha^T x} = \max_{y \in D} \min_{i \in [d]} \frac{y_i}{\alpha_i}.$$

*The only change would be in the first step; using the same rescaling idea ($z_x = \frac{x}{\alpha^T x} \in \mathbb{R}_+^d$), one could prove that*

$$\inf_{x \in \mathbb{R}_+^d} \max_{y \in D} \frac{y^T x}{\alpha^T x} \geq \inf_{\substack{z \in \mathbb{R}_+^d \\ \alpha^T z = 1}} \max_{y \in D} y^T z,$$

*while the reverse inequality trivially holds since $\left\{ z \in \mathbb{R}_+^d \,|\, \alpha^T z = 1 \right\} \subset \mathbb{R}_+^d$. The rest of the proof follows without any change.*

*Notably, since this lemma is used in all our proofs to calculate the CR for the worst-case reward expectations, it implies that we would get the same results were we to define the CR as $CR^L(P) = \inf_{r_h \in \mathbb{R}_+^{SA}} CR^L(P, r)$.*

$\square$

**Lemma 2.** *Let $d \in \mathbb{N}$. Also, let $D \in \mathbb{R}_+^d$ be a convex compact set and $\mathcal{P} \in \mathbb{R}_+^d$ be a convex compact polytope, both assumed to be nonempty. Then*

$$\inf_{\alpha \in \mathcal{P}} \max_{y \in D} \min_{i \in [d]} \frac{y_i}{\alpha_i} = \min_{\alpha \in \mathcal{P}} \max_{y \in D} \min_{i \in [d]} \frac{y_i}{\alpha_i},$$

*where we define all ratios to be $+\infty$ if the denominator equals zero.*

*Proof.* We assume w.l.o.g. that $\mathcal{P} \neq \{0\}$, since the infimum over a singleton is always equal to the minimum (in this case, both equal $+\infty$), and the result trivially holds.

Next, for all $\alpha \in \mathcal{P}$, define $f(\alpha) = \max_{y \in D} \min_{i \in [d]} \frac{y_i}{\alpha_i}$. Notice that for any $\alpha \in \mathcal{P}$ s.t. $\alpha \neq 0$, there exists $i_0 \in [d]$ such that $\alpha_{i_0} > 0$, and so

$$f(\alpha) = \max_{y \in D} \min_{i \in [d]} \frac{y_i}{\alpha_i} \leq \max_{y \in D} \frac{y_{i_0}}{\alpha_{i_0}} < \infty,$$

where the last inequality follows from the compactness of $D$. In particular, since $\mathcal{P} \neq \{0\}$ and is nonempty, such $\bar{\alpha} \neq 0$ exists, and thus $\inf_{\alpha \in \mathcal{P}} f(\alpha) \leq f(\bar{\alpha}) < \infty$, so the value at the optimization problem in the l.h.s. is finite.

We next prove that $f(\alpha)$ is quasi-concave over $\mathcal{P}$, namely

$$\forall \alpha \neq \beta \in \mathcal{P}, \lambda \in (0,1): \quad f(\lambda\alpha + (1-\lambda)\beta) \geq \min\{f(\alpha), f(\beta)\}.$$

First, if $\alpha \neq 0$ and $\beta = 0$ (or the opposite), for any $\lambda \in (0,1)$ we have that

$$f(\lambda\alpha + (1-\lambda)\beta) = f(\lambda\alpha) = \frac{1}{\lambda} f(\alpha) \geq f(\alpha) = \min\{f(\alpha), f(\beta)\},$$

where we used the non-negativity of $f(\alpha)$ and the convention that $f(0) = +\infty$. Next, assume that both $\alpha, \beta \neq 0$. Also, let $y^\alpha$ such that

$$y^\alpha \in \arg\max_{y \in D} \min_{i \in [d]} \frac{y_i}{\alpha_i}$$

and similarly define $y^\beta$. Such $y$ must exist, since we could always write

$$f(\alpha) = \max_{y \in D} \min_{i \in [d]} \frac{y_i}{\alpha_i} = \max_{y \in D} \min_{i \in [d]: \alpha_i > 0} \frac{y_i}{\alpha_i}.$$

The maximum over a finite number of linear functions is continuous and the set $D$ is compact, so a maximizer in $D$ is always attainable. Using these definitions, we have,

$$\begin{aligned}
f(\lambda\alpha + (1-\lambda)\beta) &= \max_{y \in D} \min_{i \in [d]} \frac{y_i}{\lambda\alpha_i + (1-\lambda)\beta_i} \\
&\geq \min_{i \in [d]} \frac{\lambda y_i^\alpha + (1-\lambda)y_i^\beta}{\lambda\alpha_i + (1-\lambda)\beta_i} && (D \text{ is convex}) \\
&\overset{(*)}{\geq} \min_{i \in [d]} \min\left\{\frac{y_i^\alpha}{\alpha_i}, \frac{y_i^\beta}{\beta_i}\right\} \\
&= \min\left\{\min_{i \in [d]} \frac{y_i^\alpha}{\alpha_i}, \min_{i \in [d]} \frac{y_i^\beta}{\beta_i}\right\} \\
&= \min\{f(\alpha), f(\beta)\}.
\end{aligned}$$

Relation $(*)$ is due to the inequality $\frac{a+c}{b+d} \geq \min\left\{\frac{a}{b}, \frac{c}{d}\right\}$ for $a, b, c, d \geq 0$, and one could easily verify that the inequality is still valid when either $b = 0$ or $d = 0$.

Finally, recall that $\mathcal{P}$ is a compact convex polytope; in particular, each interior point could be represented as a convex combination of one of its finite extreme points $ext(\mathcal{P})$. Then, by the quasi-concavity, the value of each interior point is lower-bounded by the value of at least one of these extreme points so that

$$\inf_{\alpha \in \mathcal{P}} f(\alpha) = \min_{\alpha \in ext(\mathcal{P})} f(\alpha).$$

This proves that the infimum is attainable by a point in $\mathcal{P}$, thus concluding the proof. $\qquad \square$

