# OpenReview forum: "The Value of Reward Lookahead in Reinforcement Learning"
_NeurIPS.cc/2024/Conference — NeurIPS 2024 spotlight_

### Official Review · Reviewer_C3mL · 2024-07-07

**Soundness:** 4
**Presentation:** 2
**Contribution:** 2
**Rating:** 7
**Confidence:** 3

**Summary:**

The paper investigates theoretically the advantage that RL agents get from reward lookahead. In a tabular MDP setup, the reward is postulated as being a random variable $R_h(s, a)$, whose value is, by default, revealed to the agent after taking the action $a$ in state $s$ at a time-step $h$. Authors calculate exactly/prove bounds on the ratio between return obtained by a no-lookahead agent and an agent that has access to information about the particular values (samples) of the reward $L$ steps ahead. Exact calculation is given in the two cases of (1) fixed rewards' expectations, fixed dynamics (2) worst-case expectations, fixed dynamics, and the (tight) bounds are given the case of (3) worst-case expectations and worst-case dynamics. Authors also discuss the connection of their results to the notion of coverability coefficient, and walk the reader through several examples of MDPs including chain-, grid- and tree-shaped environments.

**Strengths:**

The paper presents the main theorems rigorously. It provides extensive and very detailed proofs for all of its statements in the appendix. I also liked the idea of discussing proof sketches in the main part of the paper. The paper discusses the connections to the existing literature in detail, and carefully spells out the differences to previous approaches.

**Weaknesses:**

The presentation of the paper could be improved in my opinion. The paper briefly mentions several motivating examples in the introduction (trading with known prices, ride-sharing, traffic control), but this did not give me a good understanding of what the problem is - I only understood this once I got to the formal statements in section 2.1. It would be good to have more clearly and comprehensively presented "central example" that would show exactly what is going on.

The paper also suffers from moving important bits to appendix - such as Figure 3, which quite important to understand Part III of the proof sketch for Thm 1. The treatment of the Part II proof is also quite brief (although the main idea of using minimax theorem for the policy maximisation/reward minimisation is quite clear).

I did not get a clear take-away from the paper: although the results were mathematically meaningful, I did not get a sense of how much should I care about reward lookahead. Looking at those pathological edge cases of reward distributions seems unavoidable from the technical point of view, but maybe could be mitigated by e.g. examples constraining the distributions to be some reasonable family. Since the contribution itself relies on a quite straightforward proof idea (c.f. Q1 below), the strength of the paper stems mostly from meticulously working through the details of extending this to a broader setup. It would be good to see an example where the theorem helps to resolve some more-than-toy problem.

**Questions:**

1. The central idea of using tree-shaped MDPs plus long-shot reward distributions with vanishing probability $\epsilon$ and then moving $\epsilon \to 0$ is quite straightforward. The value of the paper therefore seems to be in extending this idea to cover the case of $L < H$ and various other edge cases. However, the complications in formulas in Thm 1/3 seem to stem from the problem that there's "too many"/"too few" states wrt to the time horizon (which is mitigated by adding the waiting action). Wouldn't it simplify at lot if you would just index the states and actions by $h$, which would make the MDP always a complete $A$-ary tree?
2. All of the development is done in a finite-horizon case. However, you also mention using the receding horizon idea - would that make it possible to extend your work to get around this limitation?
3. What would be the value/an application of this work in a more-than-toy setup?

**Limitations:**

Yes.

---

> ### Author Rebuttal · Authors · 2024-08-02
>
> We thank the reviewer for the comments and apologize for any clarity issue; we will consider adding a central example to improve the clarity of the introduction.
>
> Sadly, due to the page limit, we had to move to the appendix some parts of the proof that we also find essential - we intend to use all the extra space in the camera-ready version to both address the remarks in the different reviews and then move proof sketches/figures to the main text of the paper. As space permits, we will prioritize the parts suggested by the reviewer.
>
> We thank the reviewer for the opportunity to highlight our contributions. Reward lookahead is a different feedback paradigm than the standard one in RL that covers many realistic settings. This includes the trading/traffic examples in the people, but also other situations - for example, if the rewards depend on the weather forecast (e.g., in electricity grids).  In some situations, it naturally exists, while in others, agents must actively pay to gather this information. The goal of our work is to establish a deep understanding of the effect of this information on the value. This has a direct implication when information has a price, but even when it is free - algorithms that incorporate lookahead are naturally much more complex and are oftentimes tailored to each specific application.  It is thus important to ask how much do we expect to gain from using this information before deciding whether it is worth the direct and indirect costs.
>
> While we agree that the environment that yields worst-case behavior is somewhat intuitive, we emphasize that our results also give a tight characterization as a function of the dynamics of the environment - we believe that these results are not straightforward (as we elaborate in the answer to question 1). In particular, if we are interested in the CR for a specific environment, the ratio $CR(P,r)$ can easily be calculated using standard planning modules while $CR(P)$ forms a zero-sum game between two Markov policies. Our results also draw a surprising link between reward lookahead and coverability/other concentrability coefficients; in particular, to our knowledge, it is the first instance where coverability appears intrinsically and not as an assumption/constant in upper bounds. As a final note, the definition of the CR is invariant to scaling, and since long-shot distributions are bounded, the same ratios $CR^L(P)$ and $CR^L$ would also be obtained when only considering Bernoulli rewards. Thus, to get different ratios, it is not enough to limit ourselves to ‘standard’ distributions, but we might also need additional regularity conditions.
>
> # Questions:
>
> 1. We agree that for delayed trees, longshot rewards are an intuitive choice. However, we would like to emphasize that we prove that longshot rewards are the worst-case for *any* dynamics and/or expected rewards, a result that we find unintuitive. This includes dense environments such as contextual bandits and situations with tradeoffs between navigation and reward collection such as in the chain (prophet)/grid examples. We also believe that the derivation of the closed-form expression of $CR(P)$, though relying on well-known tools such as the minimax theorem, was non-trivial to devise.
> *Delayed tree and non-stationary environments:* as a remark, we intentionally chose to present a worst-case environment with stationary dynamics – otherwise, one could legitimately claim that the $H$ dependence might be due to the non-stationarity, and a tighter bound of $\approx SA$ could be the ‘right’ CR in stationary environments. Moreover, even in non-stationary environments, we believe that a similar loop mechanism is still necessary. The idea is to create an environment where only one reward could be collected (to minimize the value $V^0$), but the probability of collecting this reward greatly increases when lookahead information is available. For the first part - the states where a reward could be collected must be transient (so that no-lookahead agents would not be able to go back there), while for the second part, we need to allow the lookahead agent to decide when to collect the reward based on its observed information – so the environment requires some waiting mechanism.
>
> 2. While we did not study it in this paper, we believe that some of our techniques could be extended to discounted situations, and maybe (under some regularity conditions) to stochastic shortest paths. We leave this for future work. For infinite-horizon average reward, it is not too hard to prove that the CR can approach zero even for environments with a constant number of states (by controlling the effective horizon via loops), but it is still interesting to analyze the CR as a function of the dynamics.
>
> 3. As previously mentioned, one could take the closed-form expressions in the theorems and evaluate them on non-toy applications that have lookahead information. In the paper itself, we aimed to give closed-form results and intuition, so smaller examples were more natural. In particular, the examples studied in the paper give valuable insights into different properties of environments and their effect on the value of lookahead information (reward density, navigation elements, etc.)
> We also believe that the connection to coverability\concentrability could lead to additional theoretical applications - in any situation where coverability appears, one could reformulate the problem using the ratio between lookahead values as an analysis tool.

---

> > ### Comment · Reviewer_C3mL · 2024-08-12
> > **Response**
> >
> > I thank the authors for their response.
> >
> > I don't think I agree with the un-intuitiveness of long-shot rewards constituting the worst-case wrt dynamics and reward expectations - that is, reading the paper, I expected exactly those distributions to play the central role before getting to Definition 2. (I would recommend remarking on this fact somewhere in the introduction). On the other hand, I do agree that devising the proof seems non-trivial.
> >
> > I liked the framing of the result as "how much should an agent pay for access to the future information" - I don't think it was present in the paper, and, although logically trivial, it still made me think of the result in a different light - I would recommend working it into the paper.
> >
> > I decided to keep my current positive rating.

---

### Official Review · Reviewer_ofoH · 2024-07-14

**Soundness:** 3
**Presentation:** 3
**Contribution:** 3
**Rating:** 6
**Confidence:** 3

**Summary:**

This paper examines the value of having lookahead information about future rewards in reinforcement learning. Specifically, it analyzes the competitive ratio between the optimal agent under no lookahead versus agents that can see reward realizations for some number of future timesteps. This competitive ratio is defined as the value (expected cumulative reward) of a standard RL agent with no lookahead divided by the value of an agent with L-step lookahead. Using this measure the authors provide tight bounds on this competitive ratio for different lookahead ranges, characterizing the worst-case reward distributions and environments.

Interestingly, they show connections between their results and fundamental quantities in offline reinforcement learning and reward-free exploration. The analysis provides theoretical insights into the value of future reward information, and opens up the roadmap for future works on transition look-ahead and development of approximate planning algorithms.

**Strengths:**

**Strengths**

- The paper provides a rigorous theoretical analysis of the value of future reward information in RL, covering the full spectrum from one-step to full lookahead. It derives tight bounds on the competitive ratio for various lookahead ranges, characterizing worst-case reward distributions and environments. Notably, the competitive ratio is shown to be closely related to concentrability coefficients used in offline RL and reward-free exploration, suggesting a deeper connection between these areas.

- The analysis also includes specific environment types (e.g., chain MDPs, grid MDPs) to provide concrete examples. Notably they introduce  "delayed tree" environment that exhibits near-worst-case competitive ratios, offering insights into what makes lookahead challenging to utilize.

- The focus on worst-case scenarios provides robust guarantees and insights that complement average-case analyses common in RL literature. This approach lays crucial groundwork for understanding lookahead in more complex environments and could inform future practical algorithm design, bridging theoretical robustness with potential real-world applications.

**Weaknesses:**

**Weakness**

- The analysis assumes perfect knowledge of future rewards, which may be unrealistic in many practical scenarios where only noisy or partial information might be available.

- The paper focuses on theoretical analysis in a simplified tabular setting, which may not directly translate to more complex real-world RL problems or environments. Experiments on approximate planning for complex environments would complement its theoretical results and could have provided additional insights or validation.

**Questions:**

-

**Limitations:**

-

---

> ### Author Rebuttal · Authors · 2024-08-02
>
> We thank the reviewer for the feedback.
>
> * **Perfect future information:** as stated in the conclusions section, we agree that situations with noisy/imperfect predictions are of great interest and should be further investigated in future work. Nonetheless, we believe that the case of perfect information is still important to study, due to multiple reasons:
>
> 1. *Applicability:* in some problems, perfect or near-perfect information is available. For example, consider an inventory management problem, where supplies are bought in a market. The item prices are exactly known before each transaction, even if the market itself is stochastic - so this scenario could be formulated as one-step reward lookahead. Another scenario is ride-sharing: assume we travel between two points but are willing to pick up other travelers on the way. The knowledge of where and when travelers want to be picked from is (approximately) accurate.
>
> 2. *A stepping stone towards predictions.* The case of perfect information is the edge case in many different formulations of reward predictions. One formulation is when predictions become increasingly noisy as we look further into the future; our results are the limit case of no-noise up to a certain point, and infinite noise later on. Another situation is when predictions can either be perfect or adversarial, and agents need to learn how to utilize the prediction without losing too much if they are inaccurate (‘consistency-robustness tradeoff’); our paper analyzes the ‘consistent’ case. In both cases, it would be extremely hard to analyze the general case without having tight characteristics at the limit of perfect information.
>
> * **Lookahead in complex environments:** we also think that extending our analysis beyond tabular environments is very interesting and could have numerous practical implications. Yet, we would also like to stress that even in the tabular setting, there are many interesting open questions: planning with multi-step lookahead information, learning when lookahead information is available, tight characterization of transition lookahead and more. We believe that before moving to more complex settings, it would be beneficial to establish a deeper understanding of lookahead in tabular settings.

---

### Official Review · Reviewer_mGpv · 2024-07-14

**Soundness:** 3
**Presentation:** 4
**Contribution:** 4
**Rating:** 7
**Confidence:** 2

**Summary:**

This paper aims to quantifiably analyze the the value of future reward lookahead in Reinforcement Learning settings where future reward information is available before-hand. The authors utilize competitive analysis, and characterize the worst-case reward distribution while also deriving exact ratios for the worst case reward expectations between standard RL agents and agents with partial future-reward lookahead information.

**Strengths:**

1) The paper provides an important theoretical study shedding light on the importance of future reward lookahead in Reinforcement Learning settings. The problem is very relevant to not only simulation-based but also real-world scenarios where future reward information will either be known or can be inferred via exploration.
2) The paper is well-written, easy to follow and provides useful intuitions throughout allowing for readers to gain insight into the problem and the theoretical analysis presented.

**Weaknesses:**

1) There seems to be an important missing piece, specifically in situating the work with respect to the existing literature. A rich literature exists in the field of Control Theory that talks about the rollout approach and there has been theoretical evidence shedding light on advantages of rollout, which assumes the presence of future rewards (or <state,action> Q-values). How does this work connect to the Rollout approach and/or its variants?
2) It would be helpful to have conclusive statements along with definitions presented in the work. For example, line 121 - what are the implications of the fact that "the competitive ratio is the worst-possible multiplicative loss of the standard (no-lookahead) policy" to policy learning?
3) There seems to be some confusion about how dense rewards have been defined in the paper, line 344 onwards. Do the authors make the assumption that a reward is available in every state? How is the density defined in this case? Also, it is not clear why it is important to consider all states with non-zero rewards. With sufficient number of trials given sparse rewards, agents could still navigate to rewarding future states. How does this break the competitive ratio analysis further presented in the paper?

**Questions:**

Please see weaknesses section for questions.

**Limitations:**

While authors include useful examples towards the end of the work, it will be useful to include a section on Broader Impact as it may allow to see how the results from this work can be translated to empirical studies.

---

> ### Author Rebuttal · Authors · 2024-08-02
>
> We thank the reviewer for the insightful comments.
>
> 1. Thanks for the comment. To the best of our knowledge, there are two types of rollout approaches that are applied in control/RL: i) Rollout as a tool to perform planning: in this case, it is a computational scheme, and no future information is revealed. This kind of rollout is less relevant to our work (even though it might later be used as a planning tool for multi-step lookahead). In particular, ‘standard’ Q-values at the leaf of a rollout usually fall under this case. ii) Rollouts with additional information about state-reward realization: to our knowledge, this case is a particular instance of the Model Predictive Control Paradigm. We will try to clarify our discussion to reflect this, and will be happy to hear about any specific reference that the reviewer thinks we should discuss.
>
> 2. The no-lookahead agent is the standard agent used throughout the RL literature and serves as the ‘off-the-shelf’ agent. Therefore, the competitive ratio in our paper quantifies the maximal potential gain when moving from the standard RL scenario to agents that utilize future reward information. For example, consider a situation where we might get lookhead information, but obtaining it has a price (either because the information itself is costly or just because the lookahead algorithm is much more complicated). The CR can help determine whether the potential gain is worth the price. Another application is RL with adversarial rewards - as mentioned in Remark 1, our CR is an upper bound on the best achievable CR in adversarial settings. In fact, for full lookahead, one could calculate the policy that optimally covers the space (see, e.g., Al-Marjani et. al, 2023), and our results imply that it achieves the optimal CR with adversarial rewards. Finally, our results provide interesting insights on sequential decision-making and MDPs. In particular, our results provide a new definition of the coverability coefficient that goes beyond the mathematical expression: the worst case CR between vs. full lookahead given the dynamics. To our knowledge, this is the first result that obtains the coverability coefficient as an intrinsic quantity - previous papers only rely on it for the analysis or obtain upper bounds that depend on it. We will further discuss this in the final version of the paper.
>
> 3. We apologize for the confusion. When we say that rewards are dense, we assume that if a reward could be obtained for one action in some state, then the expected rewards of all other actions of this state can be at most $C$ times lower. If all the expected rewards at some state are zero, then the rewards will deterministically be equal to zero (since the rewards are non-negative), and such states do not affect this result. When this assumption holds, the cost of balancing immediate reward collection and future reward collection is bounded - even if we focus on future rewards, we still collect a fraction of the best immediate reward. This is not the case, for example, in the prophet problem - we either collect immediate rewards and move to non-rewarding states or collect a zero immediate reward and move to a state with a positive value. We will rephrase this example to make it clearer in the final version of the paper. We are also open to suggestions on alternative names for this scenario.
>
> 4. We completely agree that taking this work to practical settings is an important future work - we will discuss it in the final version of the paper.

---

> > ### Comment · Reviewer_mGpv · 2024-08-11
> > **Response to Author Rebuttal**
> >
> > Thanks for your response! I will maintain my assessment of the paper.

---

### Decision · Program_Chairs · 2024-09-25

**Decision:**

Accept (spotlight)

**Comment:**

The reviewers unanimously agreed on the importance and value of the contributions of this paper. Two reviewers praised the paper for its clarity, and all agreed about the high technical rigor. Based on the discussion, no further issues were uncovered, and I believe this is a clear accept.